# RIG: Synergizing Reasoning and Imagination in End-to-End Generalist Policy

**Zhonghan Zhao**[1,2*]    **Wenwei Zhang**[2*]    **Haian Huang**[2]    **Kuikun Liu**[2]    **Jianfei Gao**[2]
**Gaoang Wang**[1†]    **Kai Chen**[2†]
[1]Zhejiang University    [2]Shanghai AI Laboratory
{zhonghan.22, gaoangwang}@intl.zju.edu.cn, {zhangwenwei,chenkai}@pjlab.org.cn
[*]Equal contribution.    [†]Corresponding author.

## Abstract

Reasoning before action and imagining potential outcomes (*i.e.*, world models) are essential for embodied agents operating in complex open-world environments. Yet, prior work either incorporates only one of these abilities in an end-to-end agent or integrates multiple specialized models into an agent system, limiting the learning efficiency and generalization of the policy. Thus, this paper makes the first attempt to synergize **R**easoning and **I**magination in an end-to-end **G**eneralist policy, termed **RIG**. To train RIG in an end-to-end manner, we construct a data pipeline that progressively integrates and enriches the content of imagination and reasoning in the trajectories collected from existing agents. The joint learning of reasoning and next image generation explicitly models the inherent correlation between reasoning, action, and dynamics of environments. It thus exhibits more than $17\times$ sample efficiency improvements and generalization in comparison with previous works. During inference, RIG first reasons about the next action, produces potential action, and then predicts the action outcomes, which offers the agent a chance to review and self-correct based on the imagination before taking real actions. Experimental results show that the synergy of reasoning and imagination not only improves the robustness, generalization, and interoperability of generalist policy but also enables test-time scaling to enhance overall performance.

## 1 Introduction

To navigate the complexities of open-world environments, two quintessential human faculties are *de facto* to embodied agents: imagination of prospective outcomes and reasoning. Although reasoning endows agents with the ability to deconstruct task objectives into executable plans through logical inference, it inherently operates within the constraints of perceptual history. This limitation underscores the complementarity of world models that learn the environmental dynamics, which not only allows the agent to predict action consequences but also facilitates risk-aware decision-making by evaluating hypothetical trajectories.

The synergistic integration of reasoning and imagination constitutes an indispensable foundation for more intelligent and robust embodied agents operating in dynamically evolving environments. However, these two abilities are typically implemented in separate models. Specifically, reasoning primarily exists in large vision language models (VLMs) (Zhao et al., 2023; Wang et al., 2023a;b; Zhao et al., 2024a) that parse visual input and produce textual insights and actions (Figure 1(a)), which lack explicit mechanisms for future prediction. In contrast, world models (Lin et al., 2023; Hafner et al., 2023) specialize in predicting future frames from video data (Figure 1(b)), which suffer from data inefficiency due to the implicit learning of concepts, physical laws, and environment dynamics. Recent attempts (Zhou et al., 2024b; Zhang et al., 2023; Zhao et al., 2024b) combine reasoning and imagination by connecting VLMs and visual generative models (VGMs). Yet, the integrated system (Figure 1(c)) prevents end-to-end optimization of the agent, leaving the mutual benefits between reasoning and world models underexplored.

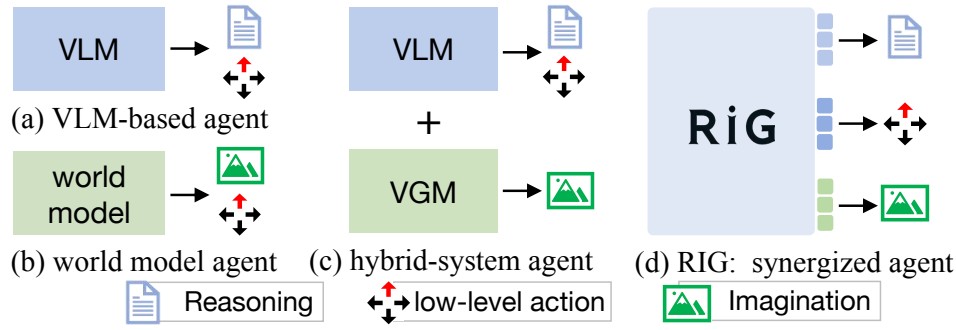

Figure 1: **Comparison between conventional agents and RIG.** RIG produces reasoning, actions, and imagination within a single Transformer.

To bridge these gaps, this paper makes the first attempt to synergize **R**easoning and **I**magination in an end-to-end **G**enralist policy, termed **RIG** ( Figure 1(d)), RIG learns textual reasoning, low-level action control, and image generation through the sequence-to-sequence modeling objective within an autoregressive Transformer, as we hypothesize that the explicit modeling of the logic and motivation behind actions and their consequences could make RIG capture open-world dynamics more comprehensively and improve the sample efficiency of training.

We develop RIG by adopting a progressive data collection strategy because existing datasets typically lack trajectories that contain interleaved image observations, precise actions, and high-quality textual reasoning. Based on initial trajectories collected from humans (Fan et al., 2022) and existing agents (Lifshitz et al., 2023) that contain only actions and image frames, we first use VLM to insert textual rationales before each action on the trajectory and train RIG-*basic* with the reasoning-enriched trajectories. During inference, RIG-*basic* generates actions purely from textual and visual inputs, without leveraging imagined future frames, as decisions are executed immediately based on current observations.

To further leverage visual imagination in reasoning to further improve the robustness of the policy, we collect unsuccessful trajectories from RIG-*basic* and adopt GPT-4o to review and revise these trajectories. Then, the suboptimal trajectories are taken as dreamed trajectories and combined with their corresponding revisions to form dream-review style trajectories for training RIG (also noted as RIG-*lookahead* for clarity). In contrast to RIG-*basic* that conduct reasoning *without* imagination, RIG-*lookahead* learns to first generate a trajectory by taking the predicted images as the environment states, and then review the hypothetical trajectory in reasoning, and predict revised action that changes the environment. Such a design provides scalability at inference time, where the number of steps in the dream trajectory can be scaled so that the agent can more comprehensively understand the effectiveness of the action and make future-aware decisions.

We extensively evaluate RIG in the diverse, open-world Minecraft environment. Experimental results show that RIG upgrades the state-of-the-art results on embodied tasks, image generation, and reasoning benchmarks by $3.29\times$, $2.42\times$, and $1.33\times$, respectively. Such a superior performance is achieved by training RIG on only **111 hours** of videos, which is **17**$\times$ fewer than previous works that rely on 2000 hours of videos. Moreover, when scaling the training data, environmental interactions, and the lookahead steps during reasoning, the generalization ability and robustness of RIG consistently improve, which implies the potential of synergizing reasoning and imagination in embodied agents. Our main contributions are summarized as follows:

- We introduce an end-to-end generalist policy that synergistically integrates explicit embodied reasoning and visual imagination.

- We propose a progressive data collection strategy coupled with straightforward language model-based training to efficiently implement our method.

- Our method naturally supports test-time scaling, enabling dynamic lookahead reasoning that enhances action robustness and reduces trial-and-error during inference.

## 2 RELATED WORK

**Embodied Agents in Minecraft.** Minecraft presents a significantly open-ended and complex environment (Johnson et al., 2016; Guss et al., 2019; Fan et al., 2022; Wang et al., 2023c; Cai et al., 2023a) for embodied agents. Early approaches leveraged explicit world models to predict future states (Hafner et al., 2023; Cai et al., 2023b) but lack textual reasoning capabilities. Inspired by large language models (LLMs) (Brown et al., 2020; Touvron et al., 2023), subsequent methods combined LLMs with low-level controllers to address long-horizon tasks. For example, Voyager (Wang et al., 2023a) and STEVE (Zhao et al., 2023) used LLMs for high-level planning integrated with code databases, while others like Jarvis-1 (Wang et al., 2023b) paired LLMs with pre-trained low-level policy models such as VPT (Baker et al., 2022). However, these methods typically lack a world model to explicitly anticipate future visual outcomes. More recently, MineDreamer (Zhou et al., 2024b) integrates a world model and a policy controller, yet treats vision generation and policy control as separate modules, limiting coherent multi-modal reasoning. In contrast, RIG first attempts to explore an end-to-end generalist policy that simultaneously learns textual reasoning, visual imagination, and low-level action predictions to achieve high generalization ability and sample efficiency.

**World Models for Embodied Agents.** Learning robust world models is essential for embodied agents to effectively plan and act within simulated environments (Oh et al., 2015; Kaiser et al., 2020). Early approaches primarily focused on action-conditioned video prediction or latent imagination for sample-efficient rollouts (Hafner et al., 2020; 2021; Schrittwieser et al., 2020; Hansen et al., 2022; Lin et al., 2023), yet they often tightly coupled the world model with specific policies, limiting their adaptability. Inspired by recent successes in large-scale pre-training (Wu et al., 2023; Mendonca et al., 2023) and Transformer-based architectures (Micheli et al., 2023), several methods now leverage generalizable knowledge to model visual and textual distributions. However, these models typically overlook explicit reasoning and deeper causal relationships between actions and resulting visual states. RIG explicitly learns to model the joint distribution of textual reasoning, actions, and their visual consequences to enable more accurate predictions of complex and evolving environment dynamics.

**Unified Understanding and Generation.** Multi-modal Large Language models (MLLMs) aim to tackle understanding and generation tasks across different modalities (Lu et al., 2023; Zhou et al., 2024a) within a unified architecture. Existing methods typically train on large-scale image-text datasets to improve general visual understanding and generation capabilities (Wang et al., 2024; Yu et al., 2023; Xie et al., 2024; Zhou et al., 2024a). However, these datasets lack the interleaved action and reasoning trajectories required for training embodied agents, limiting their direct applicability to real-world embodied scenarios. Generalist policies like GATO (Reed et al., 2022) and RT-1 (Brohan et al., 2023) demonstrate multitask capabilities but optimize each task individually without fully leveraging inter-modal synergies. Our work synergizes textual reasoning, low-level action predictions, and visual generation.

## 3 METHOD

This paper makes the first attempt to explore the synergy of **R**easoning and **I**magination in an end-to-end **G**eneralist policy, termed RIG. RIG models image, textual reasoning, and textual action in a sequence-to-sequence manner (§ 3.1). We adopt a progressive data collection strategy to first obtain RIG-*basic* that can reason before action but without imagination (§ 3.2), then approach RIG-*lookahead* that learns to reason based on generated trajectories (§ 3.3).

### 3.1 PRELIMINARY

Typical generalist policies follow an autoregressive paradigm to predict actions based on observations (Hafner et al., 2023; Lin et al., 2023). RIG extends this framework by explicitly generating intermediate textual reasoning before action prediction. Specifically, given multi-modal inputs $X = \{x^{\text{IMG}}, x^{\text{TXT}}\}$ comprising visual tokens $x^{\text{IMG}}$ and textual tokens $x^{\text{TXT}}$, RIG learns to autoregressively generate textual reasoning tokens $Y$, low-level action tokens $A$, and visual prediction tokens $P$:

$$(Y, A, P) = \mathcal{F}(X), \quad X = \{x^{\text{IMG}}, x^{\text{TXT}}\}. \tag{1}$$

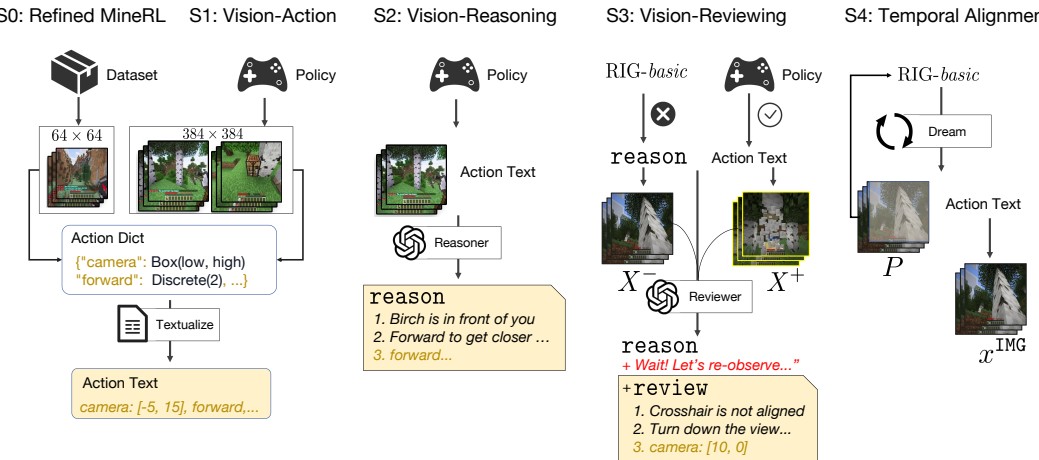

Figure 2: **Illustration of the data collection pipeline (S0–S4).** Note that at S3 (Vision-Reviewing), we run the trained RIG-*basic* and policy model (STEVE-1 (Lifshitz et al., 2023)) in parallel, keeping instances where RIG-*basic* performs poorly compared to STEVE-1.

The model is trained in an end-to-end manner using only cross-entropy loss:

$$\mathcal{L} = -\sum_{i=1} \log P_\theta(x_i \mid x_{<i}). \tag{2}$$

where $P_\theta(\cdot \mid \cdot)$ denotes the conditional probability distribution parameterized by the weights $\theta$ of RIG.

## 3.2 REASONING WITHOUT IMAGINATION

Our primary goal is to develop a synergized model capable of simultaneously generating textual reasoning, precise low-level actions, and visual outcome predictions. Since existing agents mainly produce actions, existing accessible datasets typically lack comprehensive trajectories containing all these elements. Therefore, we propose a progressive data collection strategy to gradually enrich these elements in accessible agentic trajectories. Inspired by the recent success of vision-language models (OpenAI, 2023) that can conduct chain-of-thought (CoT) reasoning given images, our first step is to add reasoning into the action-image trajectories using VLMs to obtain RIG-*basic* that can conduct reasoning before action.

**Data Collection (S0–S2).** As shown in Figure 2, we first refine or collect data from relabeled human play trajectories (S0) and specialized policies (S1), unify their formats, and add reasoning contents before each action (S2). The details are as below:

- **S0 (Refined MineRL-V0):** We use trajectories from MineRL-V0 (Guss et al., 2019) and quantize the camera actions of the original trajectory into discrete 5-degree intervals and then represent them as textual tokens. All other discrete low-level actions retain their original semantic labels.
- **S1 (Vision-Action, 446K):** We use a pretrained policy, STEVE-1 (Lifshitz et al., 2023), to collect high-resolution (384×384) image-action pairs and ensure precise visual-action alignment for learning low-level control.
- **S2 (Vision-Reasoning, 200K):** To integrate reasoning in the original trajectories, we employ GPT-4o as a **Reasoner** to annotate explicit textual rationales conditioned on visual observations $x^{\text{IMG}}$ and the corresponding low-level actions $A$, formed as $Y = \textbf{Reasoner}(x^{\text{IMG}}, A)$.

All these trajectories are rigorously filtered based on task success, diversity across environment seeds, and manual validation of reasoning quality. We train RIG-*basic* using datasets obtained from S0, S1, and S2.

**Reasoning without Imagination.** After training on datasets (S0, S1, S2), the resulting model, RIG-*basic* naturally supports multi-round interactions with the environment. As shown in Figure 3, at

Figure 3: **Inference process in RIG.** RIG follows a structured *conversation flow* through multi-turn interactions. It consistently uses the fixed word `Imagine:` to clearly separate internally imagined scenarios from real observations, thereby guiding coherent reasoning, action prediction, and visual imagination.

each step, it autoregressively generates textual CoT reasoning $Y$, low-level actions $A$, and action outcomes $P$:

$$(Y_{i+1}, A_{i+1}, P_{i+1}) \xleftarrow{\mathcal{F}} (X_i, Y_i, A_i). \tag{3}$$

This unified approach achieves significantly better generalization than traditional methods, requiring substantially fewer training samples (Figure 4).

### 3.3 LOOKAHEAD REASONING

Although RIG-*basic* demonstrates strong baseline performance, the reasoning is still purely based on the perceptual history and does not fully exploit the generative imagination capabilities. To address this, we further augment our datasets with reflective reviewing annotations in stages 3 and 4 (S3 and S4 in Figure 2), to endow the model with the ability to conduct *lookahead* reasoning, *i.e.*, internally simulate imagined trajectories first, and then take actions after reviewing the predicted future outcomes.

**Data Collection (S3–S4).** We collect reflective annotations and temporal alignment data through the following stages:

- **S3 (Vision-Reviewing, 27K):** We use a state-wise advantage filter where we only retain initial states for which STEVE-1 achieves higher expected return than the current RIG-*basic*. It avoids using a weaker policy as positive globally and ensures the positive/negative trajectories are locally comparable under identical initial conditions. We generate S3 as follows:
  - **Negative trajectory:** Generated by the previously trained RIG-*basic* model, yielding suboptimal outcomes $X^-, Y^-, A^-$.
  - **Positive trajectory:** Generated by the superior-performing policy STEVE-1 (Lifshitz et al., 2023), yielding optimal outcomes $X^+, A^+$.

We then adopt GPT-4o as a **Reviewer** to explicitly compare these parallel trajectories and generate refined reasoning: $Y^+ = \textbf{Reviewer}(X^-, Y^-, A^-, A^+)$, so that we get corrective reasoning annotations:

$$Y = \{Y^-, \text{``Wait! Let's re-observe...''}, Y^+\}. \tag{4}$$

This reflection annotation significantly enhances the ability of the model to review and correct reasoning mistakes.

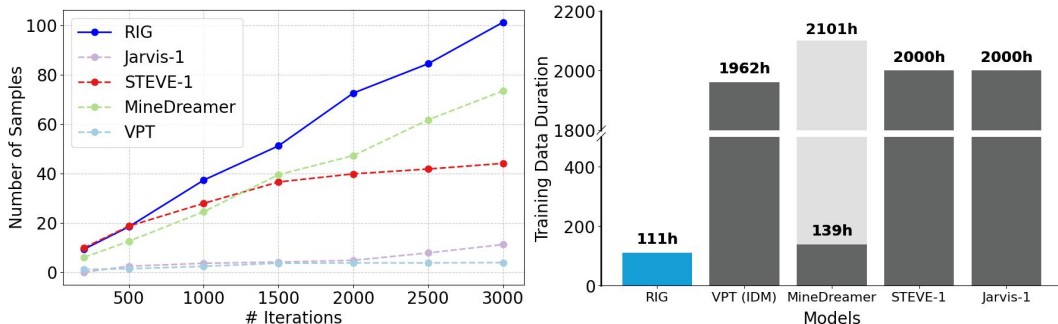

Figure 4: **Performance and data-efficiency comparison.** RIG-*basic* significantly outperforms other baselines with higher sample efficiency and achieves superior performance using only **111 hours** of training data (42h S0 MineRL-V0 and 69h S1-S4). MineDreamer (Zhou et al., 2024b), a hybrid-system model, separately trains a visual generation model (139 hours) but also relies on VPT for the policy model, increasing total data requirements. Duration of VPT (Baker et al., 2022) reflects only the IDM data used, measured as video frames, while STEVE-1 (Lifshitz et al., 2023) and Jarvis-1 (Wang et al., 2023b) also leverage the VPT dataset.

- **S4 (Temporal Alignment, 38K):** We further generate multi-step imagined visual predictions ($P$) and explicitly align them with observed ground-truth visual tokens ($x^{\text{IMG}}$) to enhance long-horizon stability: $P_{i+1} \to x_{i+1}^{\text{IMG}}$.

**Lookahead Reasoning with Imagination.** Training on datasets from stages 3 and 4 produces RIG-*lookahead*, a model that performs reasoning conditioned on imagined futures. Stage 3 employs Rejection Sampling Fine-tuning (RFT) to enhance reasoning through model-generated rollouts. We apply RFT in embodied agents by leveraging joint reasoning and visual generation, which enables self-prediction of future states, previously infeasible due to the lack of visual prediction. Only the positive trajectory $Y^+$ is optimized, while the negative $Y^-$ is excluded from loss, encouraging better self-correction.

RIG-*lookahead* simulates "dream trajectories" before acting. As shown in Figure 3, imagined steps are marked with a fixed token "`<Imagine:>`" to distinguish from observations, allowing decisions to be refined by looking $n$ steps ahead:

$$(Y_{i+1}^*, A_{i+1}^*, P_{i+1}^*) \xleftarrow{\mathcal{F}} (X_i, P_{i+1}, Y_{i+1}, ..., P_{i+n}, Y_{i+n}). \tag{5}$$

This lookahead mechanism enables internal review and correction, reducing trial-and-error interactions and enhancing decision robustness in complex embodied tasks, as shown in Figure A3.

## 4  EXPERIMENTS

We conduct comprehensive experiments to validate the effectiveness of RIG across diverse tasks, focusing on data efficiency, scalability, and the benefits of integrating generation, reasoning, and lookahead. Evaluations on embodied tasks are performed under both *Manual* (hand-only) and *Tool* (*e.g.*, iron pickaxe) to assess performance in varied embodied scenarios. More details about metrics are listed in Appendix A.3 and detailed qualitative studies are listed in Appendix A.8.

### 4.1  EXPERIMENT SETUP

**Embodied tasks.** Six tasks set up in MineRL Guss et al. (2019) (Collect: Wood/Seeds/Dirt; Explore: Dig/Explore/Tower) under *Manual* (bare-hand) and *Tool* (iron tools) settings; success and sample definitions follow Appendix A.3. Training and evaluation seeds are disjoint.

**Language & generation metrics.** Reasoning and Understanding are blind-graded; Score-Static is STEVE-21K static QA, for generation, we report FID (lower is better) and PSNR (higher is better), see Appendix A.3.

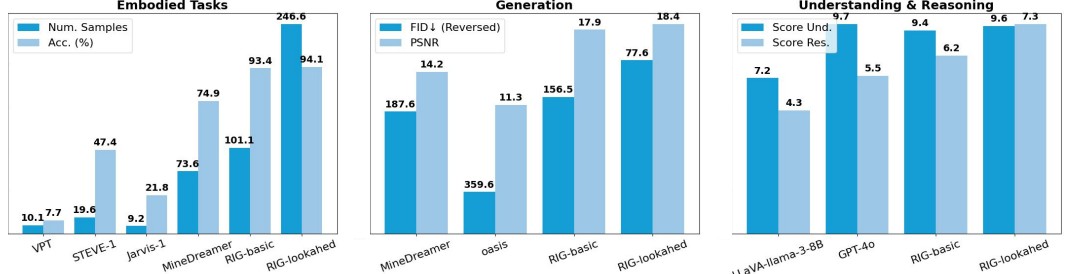

Figure 5: **Comparison across embodied, generation, and VQA/Reasoning.** RIG-*basic* equips reasoning. RIG-*lookahead* further adds reviewing.

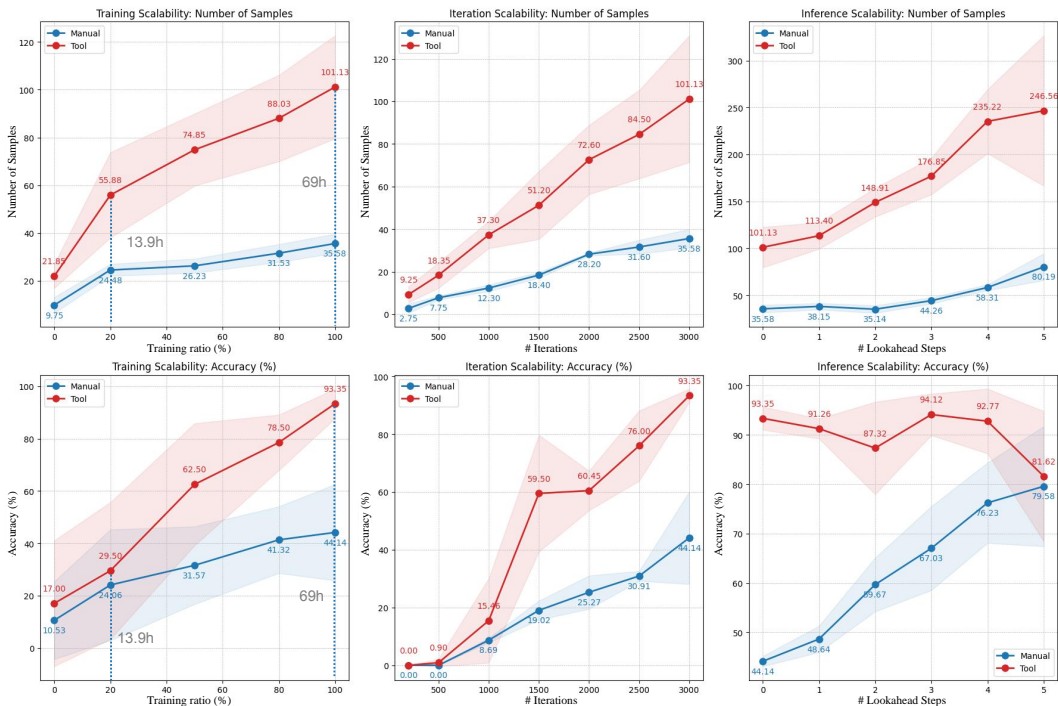

Figure 6: **Scalability over training data, iterations, and lookahead steps.** Shaded areas denote variance. The 42h MineRL-V0 pretraining is excluded from the 111h total; the training-ratio panel counts data before lookahead.

**Base model.** RIG builds on Janus-1.3B (Chen et al., 2025) with 4,096 context. SigLIP-L/16-384 (Zhai et al., 2023) encodes images; a VQ tokenizer (Sun et al., 2024) provides discrete visual IDs for the unified Transformer. Training uses sequence packing and mixed data types on XTuner-lite (Contributors, 2023).

## 4.2 DATA EFFICIENCY

As shown in Figure 4, RIG significantly surpasses other methods in terms of collected samples per iteration while requiring drastically lower total training time. Notably, RIG achieves superior performance using only **111 hours** of total data collection, considerably less than other baselines (VPT: 1962h, MineDreamer: 2101h (139h + VPT), STEVE-1 and Jarvis-1: nearly 2000h).

## 4.3 MAIN RESULTS

We first evaluate the data efficiency of RIG against prior approaches (VPT (Baker et al., 2022), STEVE-1 (Lifshitz et al., 2023), Jarvis-1 (Wang et al., 2023b), and MineDreamer (Zhou et al.,

| ID | Capabilities | | | | Number of Samples | | | | | Accuracy (%) | | | | |
|---|---|---|---|---|---|---|---|---|---|---|---|---|---|---|
| | Action | Gen. | Reason | Lookahead | wood | grass | dirt | avg. | Δ | Dig | Explore | Tower | avg. | Δ |
| *Manual (ID 0–4)* | | | | | | | | | | | | | | |
| 0 | ✓ | | | | 7.9 | 6.2 | 8.9 | 7.7 | +0.0 | 9.1 | 11.7 | 4.4 | 8.4 | +0.0 |
| 1 | ✓ | ✓ | | | 11.0 | 16.5 | 12.1 | 13.2 | +5.5 | 12.2 | 36.8 | 41.8 | 30.3 | +21.9 |
| 2 | ✓ | | ✓ | | 17.3 | 24.5 | 22.5 | 21.4 | +13.8 | 34.2 | 31.8 | 37.8 | 34.6 | +26.2 |
| 3 | ✓ | ✓ | ✓ | | 22.2 | 45.9 | 38.7 | 35.6 | +27.9 | 29.2 | 65.2 | 37.9 | 44.1 | +35.7 |
| 4 | ✓ | ✓ | ✓ | ✓ | 28.3 | 137.5 | 74.8 | 80.2 | +72.5 | 65.8 | 84.2 | 88.7 | 79.6 | +71.2 |
| *Tool (ID 0–4)* | | | | | | | | | | | | | | |
| 0 | ✓ | | | | 24.6 | 33.1 | 42.4 | 33.4 | +0.0 | 17.9 | 11.7 | 8.2 | 12.6 | +0.0 |
| 1 | ✓ | ✓ | | | 25.8 | 29.9 | 48.4 | 34.7 | +1.3 | 27.2 | 36.8 | 41.8 | 35.3 | +22.7 |
| 2 | ✓ | | ✓ | | 26.9 | 49.9 | 51.0 | 42.6 | +9.2 | 24.2 | 31.8 | 29.8 | 28.6 | +16.0 |
| 3 | ✓ | ✓ | ✓ | | 79.4 | 115.3 | 108.6 | 101.1 | +67.7 | 85.1 | 100.4 | 94.7 | 93.4 | +80.8 |
| 4 | ✓ | ✓ | ✓ | ✓ | 128.7 | 295.6 | 315.5 | 246.6 | +213.2 | 95.4 | 84.2 | 102.8 | 94.1 | +81.5 |

Table 1: **Ablation study on embodied tasks under different capability settings.** We compare different combinations of Action, Generation (Gen.), Reasoning (Reason), and Reviewing (Review). The table is divided into two groups: *Manual* (ID 0–4) and *Tool* (ID 0-4). The "Num." column represents the number of completed collecting tasks (wood, grass, dirt), while "Acc." denotes the success rate of exploration tasks. The columns "avg." is the average performance. For both metrics, we report the absolute values, along with the improvement (+$x$) **over the baseline (ID 0 for *Manual* and *Tool*).**

| ID | Capabilities | | | | Generation | | Understanding | | Reasoning |
|---|---|---|---|---|---|---|---|---|---|
| | Action | Gen. | Reason | Lookahead | FID ↓ | PSNR ↑ | Score-Stc.↑ | Score-Env.↑ | Score-Env.↑ |
| 0 | | ✓ | | | 214.5 | 16.4 | - | - | - |
| 1 | ✓ | ✓ | | | 225.6 | 16.3 | - | - | - |
| 2 | ✓ | | ✓ | | - | - | 9.0 | 7.8 | 6.1 |
| 3 | ✓ | ✓ | ✓ | | 156.5 | 17.9 | 9.4 | **8.4** | 7.3 |
| 4 | ✓ | ✓ | ✓ | ✓ | **77.6** | **18.4** | **9.6** | 8.1 | **8.5** |

Table 2: **Ablation study on Generation, Understanding, and Reasoning performance.** We compare different combinations of Action, Generation (Gen.), Reasoning (Reason), and Lookahead capabilities. "Score-Env." represents the environment-specific evaluation score from online understanding testing, while "Score-Env." denotes reasoning-specific evaluation. "Score-Stc." is computed on the static dataset STEVE-21K Zhao et al. (2023), and "FID" / "PSNR" measure image generation quality.

2024b)). We then benchmark its performance across three core task categories: Embodied Tasks, Generation Tasks, and Understanding & Reasoning Tasks. For more qualitative analysis, please refer to the case study in Appendix A.8.

**Performance in Embodied Tasks.** As shown in Figure 6, RIG-*basic* have surpassed all other baselines with **93.4%** accuracy and **101.1** collected samples. RIG-*lookahead* (extended from RIG-*basic*) produce even greater progress than RIG-*basic*, with the highest accuracy of **94.1%** and **246.6 collected samples**. For extended analysis, we conduct additional scalability and ablation studies, as shown in Figure 6 and Table 1.

**Performance in Generation and Understanding.** RIG-*lookahead* achieves the best Minecraft-style generation quality among baselines (FID **77.6**, PSNR **18.4**) (see Appendix A.8), general VQA remains on par with Janus-1.3B (see Table A5).

## 4.4 SCALABILITY

We evaluate the scalability of RIG along three key dimensions, training data ratio, iteration count, and inference steps, as illustrated in Figure 6.

**Training Scalability.** RIG exhibits strong scalability with training data volume. Increasing training data from 10% to 100% dramatically improves performance, especially at the 20% training threshold, where accuracy jumps substantially (manual: 10.53%→24.06%, tool: 17.0%→62.50%). Data diversity significantly increases at this point, allowing the agent to encounter and adapt to a broader

spectrum of complex scenarios. Beyond 20% data usage, the rate of accuracy improvement stabilizes, indicating the training paradigm reaches a steady state and that the agent's action diversity nears its upper bound. With full training data (100%), RIG achieves superior results, outperforming existing approaches such as VPT (Baker et al., 2022) and Jarvis-1 (Wang et al., 2023b) in accuracy and even surpassing STEVE-1 (Lifshitz et al., 2023) in task collection efficiency, reaching 101.13 collected tasks and 93.35% accuracy. Notably, these results are attained purely through forward reasoning, without lookahead, suggesting substantial untapped potential for further enhancement by incorporating advanced reasoning mechanisms.

**Iteration Scalability.** RIG demonstrates robust performance growth over iterations. Under standard forward inference, task collection grows consistently from 9.25 samples at iteration 200 to 101.13 at iteration 3000, particularly pronounced in tool-assisted tasks, showing rapid convergence due to effective data utilization and stable trajectory patterns. However, variance, illustrated by shaded areas, tends to increase with iterations, reflecting longer and more diverse trajectories that introduce complexity and fluctuation. Tasks involving exploration, which inherently contain more combinatorial subtasks (*e.g.*, material gathering followed by building structures), show larger variance and complexity over time. It potentially highlights the model's adaptive response to increasingly diverse scenarios.

**Inference Scalability.** The results from different lookahead steps demonstrate significant benefits from lookahead reasoning. Evaluating from the baseline at 3000 iterations, increasing steps (generating "dream trajectories") substantially improves performance. Task collection metrics exhibit rapid initial improvement and relatively low variance up to four steps, indicating accurate and stable trajectory predictions. However, variance increases at five steps, suggesting accumulated prediction errors or hallucinations become more prominent. For accuracy metrics, tool-assisted tasks maintain high performance (peaking at 94.12% at 3 steps), with a slight decrease afterward due to ceiling effects and increased prediction uncertainty. Conversely, manual tasks show consistent performance improvement through stepwise lookahead, significantly benefiting from iterative reasoning, reaching a peak of 79.58% accuracy at 5 steps.

## 4.5    ABLATIONS STUDY

**Embodied tasks.** Adding *generation* improves targeting and exploration stability; adding *reasoning* reduces redundant actions and enables goal-directed plans; adding *lookahead* yields the largest gains with minimal extra data (27K). The full model (*Action*+Gen+Reasoning+Lookahead) delivers the best sample efficiency and success (see Table 1). As shown in Table 1, the synergy of *Action*, *Generation*, *Basic Reasoning* and *Lookahead Reasoning* leads to the most robust performance, which enables structured learning and improves short-term decisions and long-horizon task completion. In the Manual setting (ID 4), there are significant gains in sample collection (from 7.7 to 80.2, +72.5) and accuracy (from 8.4% to 79.6%, +71.2). In the Tool setting, the impact is even greater, with sample counts rising from 33.4 to 246.6 (+213.2) and accuracy from 12.6% to 94.1% (+81.5), highlighting the effectiveness of the unified framework.

**Generation quality.** Action-only slightly worsens FID, but combining action with reasoning and lookahead markedly improves FID/PSNR (e.g., 156.5→77.6 FID; 17.9→18.4 PSNR), confirming that structured reasoning stabilizes the visual predictor (see Table 2). By jointly optimizing generation, reasoning, and reviewing, RIG achieves the best trade-off between action prediction, visual understanding, and environmental reasoning. The best model (ID 4) shows that lookahead reasoning enhances decision-making, improving sample efficiency and interaction robustness. This underscores the benefit of integrating multiple modalities for coherent perception and action. As shown in Table 2, enabling lookahead reasoning substantially enhances image-generation quality. RIG-*lookahead* (ID 4) attains the lowest FID (77.6) and highest PSNR (18.4), significantly surpassing variants without lookahead reasoning (*e.g.*, ID 3, FID: 156.5, PSNR: 17.9).

## 4.6    CASE STUDY

Figure 7 shows detailed case study between RIG-*lookahead* with the latest GPT-4o image generation upgraded. It illustrates that GPT-4o misjudges the distance, issues an `attack` command, and ends up "attacking air". In contrast, RIG-*lookahead* realizes the trunk is out of reach by imagination, adjusts plans, and finally succeeds. Note that we compare RIG with GPT-4o under identical prompts.

GPT-4o frequently issues invalid `attack` actions and stalls, with a valid-action rate of only 11%, while ours achieves 98%.

The training process involves S1 to S3, where imagination is predicted but used only as an auxiliary target, without a feedback loop. In S4, we apply an autoregressive method over successive imagined frames, rolling out the same action path in the real environment and training on the paired traces. This leads to RIG-*lookahead*, which produces stable predictions (Figure 7) that enhance true lookahead reasoning and improve performance. More case studies are listed in Appendix A.8

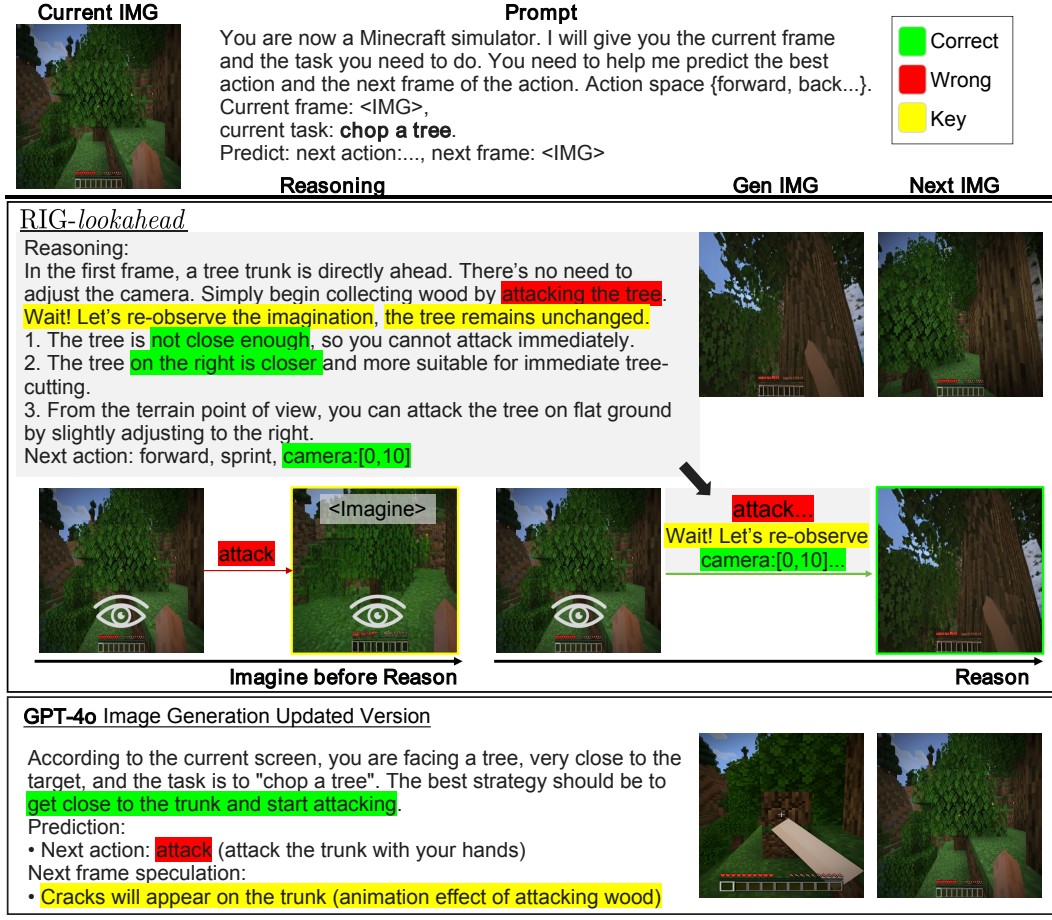

Figure 7: **Case study on lookahead ability.** This ability is crucial in embodied tasks such as fault review, helping agents accurately estimate distances and take correct actions in open-world environments.

## 5 CONCLUSION

This paper introduces RIG, an end-to-end **G**eneralist policy that integrates **R**easoning and **I**magination with superior adaptability and robustness in open-world environments. RIG unifies visual, action, and textual reasoning within a single autoregressive Transformer and can plan and reason by looking ahead with dreamt trajectories to further improve its robustness. RIG achieves new state-of-the-art performance across embodied tasks, image generation, and reasoning, with higher sample efficiency and better generalization. RIG also exhibits higher scalability with training and test-time compute. As cases shown in Appendix A.8, we hope the results of RIG could inspire future research on synergizing reasoning and imagination in embodied agents.

## ACKNOWLEDGEMENTS

We thank Jiangning Liu for insightful discussions on evaluation and for data support. We also thank Yuzhe Gu for valuable discussions and helpful comments on earlier drafts. This work was supported by the Fundamental Research Funds for the Central Universities (No. 226-2025-00167), and the National Natural Science Foundation of China (No. 62576308).

**Ethics statement.** We adhere to the ICLR Code of Ethics and confirm that all authors have read and complied with it. Our experiments are conducted entirely in a *simulated* Minecraft environment using publicly available datasets and policies (*e.g.*, MineRL-V0, STEVE-1/STEVE-21K) and model-generated rollouts. No human subjects, personally identifiable information, or sensitive attributes are collected; therefore IRB approval was not required. Reasoning annotations and blind grading use GPT-4o under dataset and API terms; we release the exact prompts, filtering rules (*e.g.*, legality weighting), and sampling criteria to document potential annotator/model bias. We discuss model limitations (*e.g.*, hallucinated "imagination" frames) and restrict claims to simulation. To mitigate dual-use risks (*e.g.*, unsafe autonomous control), we do not deploy to physical robots, provide research-only artifacts where applicable, and include safety notes in the repository. All third-party assets are cited and used within their licenses. We also report compute and data usage in the appendix and will provide an estimated carbon footprint to support environmentally responsible research.

**Reproducibility statement.** We prioritize reproducibility through clear documentation and artifact release. The training/evaluation protocol (tasks, metrics, seeds, success criteria) is specified in Appendix A.3 with full seed lists and episode counts in the appendix. Data construction S0–S4 (sources, prompts, filters, reviewer settings) is detailed in the main text and supplementary. We release the exact prompts and scripts. Model and optimization details (backbone, context length, vision encoders/tokenizers, hyperparameters) appear in §4.1 and the supplementary materials, with all configs provided as YAML files. We supply evaluation harnesses for embodied control, generation (FID/PSNR), and VQA, including Docker/Conda environment files, fixed random seeds, and commit hashes. Anonymous links in the supplemental materials include: (1) inference/training code, (2) checkpoints for *basic* and *lookahead*, (3) ablation/plot scripts to regenerate all figures/tables. Together, these materials enable end-to-end replication of results and verification of claims.

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

# A   APPENDIX

## THE USE OF LARGE LANGUAGE MODELS (LLMS)

Our work develops a multimodal LLM-based world model. Concretely, RIG builds on a 1.3B-parameter LLM backbone with unified visual–text decoding; we train and improve it using Supervised Finetuning (SFT) and Rejection Sampling Fine-Tuning (RFT) to enhance reasoning and lookahead capabilities. In addition to the backbone itself, we employ an external LLM (GPT-4o) only for technical purposes, like a reasoner and baseline in the research pipeline: generating reasoning annotations and conducting blind grading of text-only outputs under the dataset/API terms. Prompts, legality filters, and sampling criteria are documented and released to aid auditability.

Beyond these technical roles, LLMs did not play a significant part in research ideation, experimental design, or manuscript writing, the authors conceived the study, designed/evaluated experiments, analyzed results, and wrote the paper. Any automated assistance, if present, was limited to non-substantive copy-editing/LaTeX linting. The authors take full responsibility for all contents, including verifying any machine-generated intermediate artifacts, and acknowledge that LLMs are not eligible for authorship. This disclosure follows the ICLR policy on LLM usage and research integrity.

**The appendix is organized as follows:**

- **Inference Pipeline** (Appendix A.1) illustrates how RIG performs end-to-end inference: generating textual reasoning, imagined visual rollouts, and executable actions, with explicit use of the `<Imagine:>` token to support self-review and temporal consistency. Meanwhile, we compare the inference costs to STEVE-1 and MineDreamer.

- **Training Pipeline** (Appendix A.2) presents our multi-stage training framework, including offline supervised learning, GPT-4o-based reasoning and review relabeling, and imagination-grounded alignment strategies that enable lookahead-based decision-making. Meanwhile, we list the training costs of RIG.

- **Evaluation Protocol and Metrics** (Appendix A.3) presents detailed evaluation protocol and metrics.

- **Data Distribution** (Appendix A.4) analyzes the diversity of embodied tasks within our datasets, and illustrates how data volume scales with task complexity, from atomic skills like collection to composite ones like exploration and construction.

- **Component Comparison** (Appendix A.5) offers a systematic comparison with existing models, emphasizing RIG's unique capabilities in multimodal alignment, action granularity, and unified policy formulation without relying on task-specific modules.

- **Tokenizer and Base Model Selection** (Appendix A.6) explains our design choice of combining LlamaGen's VQ tokenizer with Janus as the vision-language foundation, offering a lightweight and effective setup for image-text grounding in Minecraft-like settings.

- **General VQA Benchmark Result** (Appendix A.7) compares RIG with state-of-the-arts on multimodal understanding benchmarks.

- **Qualitative Results and Case Study** (Appendix A.8) showcases examples where RIG performs internal reasoning, detects failure cases via self-review, and corrects actions before execution. We further compare RIG to GPT-4o (from reasoning and imagination comparison in Figure A5 to lookahead comparison in Figure 7), demonstrating that strong visual generation alone does not guarantee robust policy reasoning.

- **Multi-Modal Understanding Evaluation** (Appendix A.9) evaluates RIG's embodied knowledge across diverse functional categories using the STEVE-21K QA benchmark, covering survival, crafting, entity understanding, and more.

- **Multi-turn Visual Reasoning Format** (Appendix A.10) details our multi-round reasoning and imagination format, which supports fine-grained learning of vision-language-action alignment through step-by-step trajectory prediction.

- **Environment Details** (Appendix A.11) describes our experimental platform based on MineRL (Guss et al., 2019), featuring low-level egocentric control and programmable environment setup for robust and reproducible embodied evaluation.

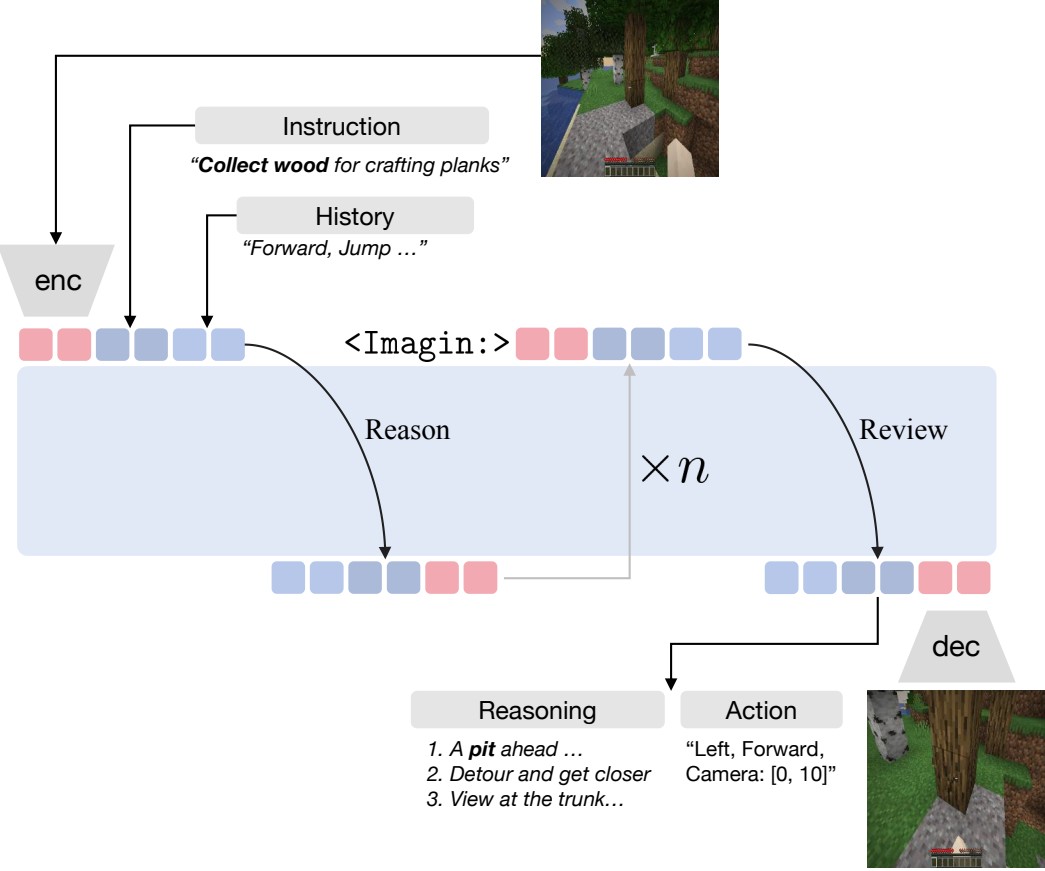

Figure A1: **Detailed inference pipeline.** RIG generates imagined visual states and corresponding reasoning to simulate multiple action trajectories, enabling self-review and corrective prediction.

## A.1 INFERENCE PIPELINE

As illustrated in Figure A1, RIG follows a multimodal autoregressive generation process. Given current observations and the task, the model produces (i) textual reasoning, (ii) low-level actions, and (iii) visual predictions of future frames. These imagined states, denoted by the fixed token `<Imagine:>`, are recursively fed back for internal reviewing and decision refinement. This mechanism allows iterative planning without environmental interaction.

**Inference costs.** For a typical control step, the policy without visual imagination (Reasoning + Action) requires approximately $2 \times 10^{12}$ FLOPs, while the full RIG pipeline with imagination (Reasoning + Action + Prediction) requires about $6.38 \times 10^{12}$ FLOPs. As summarized in Table A1, the per-step FLOPs of STEVE-1 Lifshitz et al. (2023) is about $6 \times 10^{10}$, whereas MineDreamer Zhou et al. (2024b) requires roughly $3 \times 10^{14}$ FLOPs per step ($6 \times 10^{10}$ for the controller and $3 \times 10^{14}$ for the backend image generator). Thus, among agents with an explicit world model, our synergized RIG design remains two orders of magnitude cheaper than a hybrid world-model + RL controller such as MineDreamer, while providing integrated reasoning, imagination, and lookahead. Compared to a low-level policy like STEVE-1, RIG's per-step inference cost is higher, but it offers substantially stronger performance and richer capabilities (reasoning, visual imagination, and self-review) under a $17\times$ smaller environment interaction budget.

## A.2 TRAINING PIPELINE

As illustrated in Figure A2, the training of RIG proceeds in four progressive stages:

| Method | Env. h | Imag. | Reason. | Look. | Summary | Infer FLOPs |
|--------|--------|-------|---------|-------|---------|-------------|
| STEVE-1 | $\sim$2000 | | ✓ | | Low-level policy | $6 \times 10^{10}$ |
| MineDreamer | $\sim$2000$^\dagger$ | ✓ | | ✓ | World-model + RL | $3 \times 10^{14}$ |
| RIG-basic | **111** | ✓ | ✓ | | World-model | $6 \times 10^{12}$ |
| RIG-lookahead (Ours) | **111** | ✓ | ✓ | ✓ | World-model | $6 \times 10^{12}$ |

Table A1: **Comparison between STEVE-1, MineDreamer, and RIG.** "Env. h" counts environment interaction used to train *each* agent (our 17× factor compares RIG's 111 h to STEVE-1's 2000 h). "Imag." denotes explicit visual imagination / world model, "Reason." denotes textual reasoning / review head, and "Look." denotes lookahead-based self-review. Infer FLOPs. are approximate per-step costs of inference under our setup.

- **S0/S1. Offline Supervised Fine-tuning (SFT):** The model learns to align the *dream flow* (model-generated predictions) with the *real flow* (observed data) through supervised learning. This phase improves visual state prediction quality, enhancing the accuracy of subsequent action decisions.
  *Input:* past frame, past action, task. *Output:* subtask, next action, next frame.

- **S2. Reasoning Relabeling:** A two-step process enhances decision quality. (1) An environment-based evaluator filters high-quality trajectories. (2) GPT-4o acts as a **Reviewer** to generate explicit reasoning traces and refined labels.
  *Input:* past frame, past action, task. *Output:* reasoning, next action, optionally lookahead reasoning, next frame.

- **S3. Review Relabeling:** The trained model interacts in the environment, and an evaluator filters poor trajectories. GPT-4o as a **Reviewer** analyzes the imagined traces and relabels corrections for better trajectory quality.
  *Input:* past frame, past action, task, imagined frame (`<Imagine:>`). *Output:* lookahead reasoning , corrected action, next frame.

- **S4. Temporal Alignment:** An imagined dream trajectory is generated via the autoregressive model. The entire sequence is behavior-cloned into the real environment, enabling frame-by-frame alignment and relabeling via the Reasoner.
  *Input:* dream trace (states/actions). *Output:* real visual alignment, updated reasoning annotations.

Note that RIG is trained on 111 hours of data by collecting, without reusing weights from STEVE-1 or its 2k hour corpus. We just focus solely on the final data used for training to ensure a direct metric.

Stages 0–2 are used to train RIG-*basic*, while Stages 3–4 further enhance RIG-*lookahead* with imagination-based alignment and long-horizon correction.

Eq. 4 for Stages 3-4 shows RIG-*basic* is co-trained *reasoning→action→imagination*. The **"Reasoning without Imagination"** means RIG-*basic* has not directly used imagination as input to lookahead. **Stage 3** let GPT-4o prompted to generate a coherent reasoning $Y^+$ that: (1) analyzes the failure in $A^-$, and then (2) justifies the correction leading to $A^+$. $X^-$ and $X^+$ used in Stage 3 are all observation frames, $X^-$ is a bad trial through RIG-*basic* interact with environment by action. **Stage 4** is trained on generated frame $P$ and real interaction outcome frame $X$.

Reasoning before action makes RIG globally stabler on long horizons, yet occasional local mis-actions remain. STEVE-1's greedier traces supply those hard local positives.

**Training costs.** For the training setup reported in the paper, RIG-basic (without lookahead imagination) requires about 704 GPU hours to train on 111 hours of interaction data, using 64 × A100 80GB GPUs. The additional lookahead stage for RIG-lookahead requires about 280 GPU hours on the same hardware.

A.3 EVALUATION PROTOCOL AND METRICS

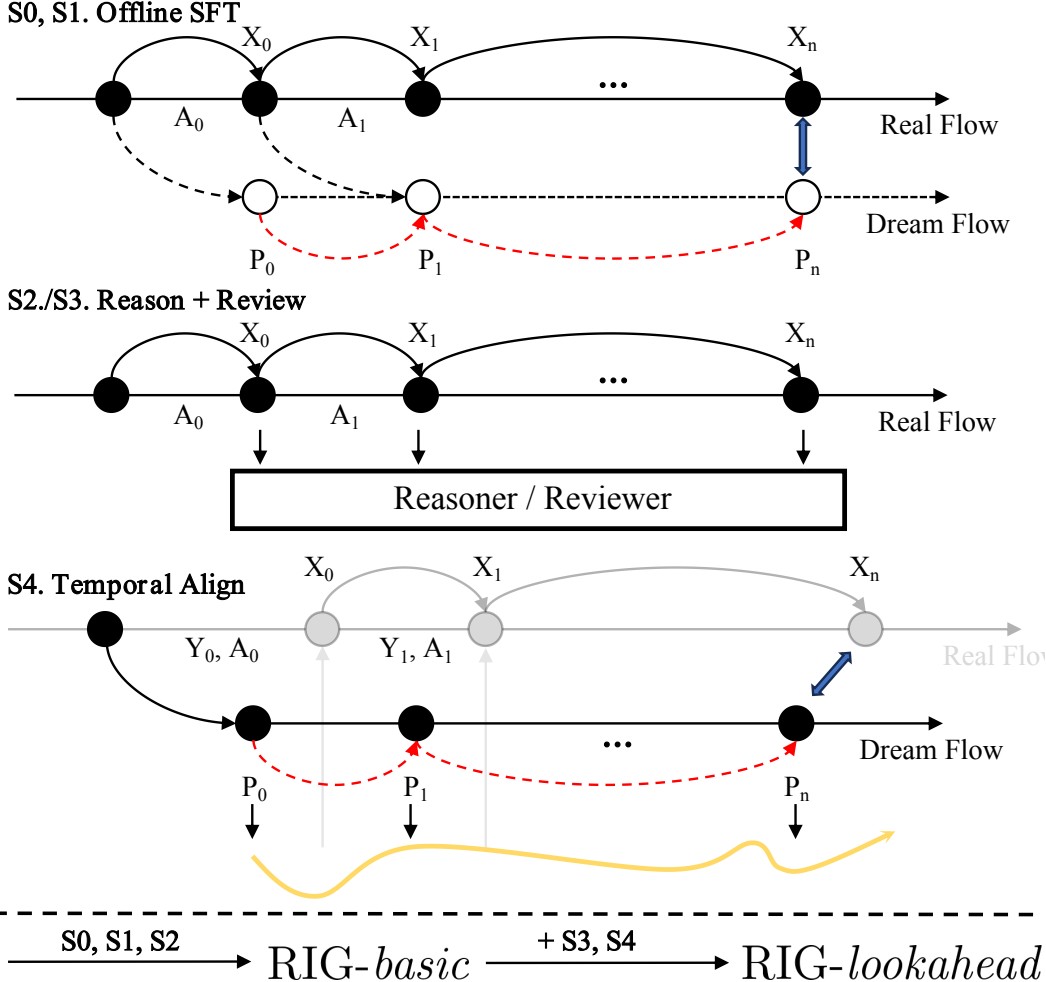

Figure A2: **Training pipeline of RIG.** S0/S1 pretrain the model by aligning real and imagined flows. S2/S3 enhance reasoning and reviewing via GPT-4o relabeling. S4 aligns temporally predicted trajectories (dream flow) with environment-grounded traces.

| Method | Vision Encoder | Parameters | Vision Quality (Gen.) | MM Quality (Und.) | Evaluations |
|---|---|---|---|---|---|
| **Autoregressive (AR)** | | | | | |
| Emu3 Wang et al. (2024) | VQ (D) | 8B | 0.68 | -0.1 | POPE, SEEDBench-Img, VQAv2 (85.2, 68.2, 75.1) |
| LlamaGen Sun et al. (2024) | VQ (D) | 111M, 343M, 775M, 1.4B, 3B | 0.68 | -0.34 | - |
| Chameleon Lu et al. (2023) | VQ (D) | 7B, 34B | 0.68 | -0.29 | - |
| Anole Chern et al. (2024) | VQ (D) | 7B | - | - | - |
| Janus Chen et al. (2025) | VQ (D) | 1.3B | 0.68 | -0.07 | POPE, VQAv2 (87, 77.3) |
| **AR + Diffusion** | | | | | |
| Show-o Xie et al. (2024) | Magvitv2 (D/C), Clip-ViT (C) | 1.3B | 0.68 | -0.15 | POPE, VQAv2 (84.5, 74.7) |
| Transfusion Zhou et al. (2024a) | VAE (C) | 0.16B, 0.37B, 0.76B, 1.4B, 7B | 0.68 | -0.01 | - |
| Fluid Fan et al. (2024) | VQ (D), VAE (C) | 369M, 665M, 1.1B, 3.1B, 10.5B | 0.68 | 0.02 | - |

Table A2: **Comparison of various unified multimodal methods**, categorized by their training approach (Autoregressive and AR + Diffusion), detailing vision encoder type, parameter scale, vision generation quality (GenEval SD3 8B), multimodal understanding quality, and evaluation benchmarks.

**Embodied tasks.** 6 tasks (Collect: Wood/Seeds/Dirt, Explore: Dig/Explore/Tower). (1) One iteration means one forward pass or frame used. (2) Num. of samples stands for the number of collected blocks. (3) Accuracy refers to the success rate of completing tasks. (4) Explore success: Dig $z \in [7, 20]$; Explore travel $\geq 300$ blocks; Tower height $\geq 15$. (5) Training seeds (*e.g.* 1-34, 701-706 for Wood) are listed in the appendix. (6) Evaluation always starts with random seeds for the random scene. (7) Tool/Manual toggles iron tools to shorten bare-hand loops.

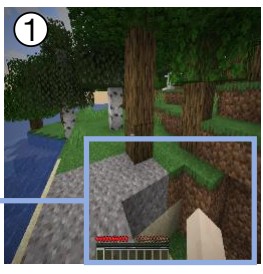

**① ② Understanding**

*You are in a forest biome surrounded by a mix of oak and birch trees. The terrain consists of dirt, and **grass** patches, with a nearby body of **water** to the left. The crosshair is positioned directly on the trunk of an oak tree...*

**③ Lookahead Reasoning**

*1. The tree trunk is already centered in the crosshair, meaning the player does not need to adjust their aim.*
*2. The player is standing on gravel, and there is a slight gap between them and the tree.*
*3. Moving forward will bring the player closer, ensuring they are within breaking range to chop the tree efficiently.*
*4. Since the tree is directly in front, there is **no need for camera adjustments** at this moment, only movement.*

*"Wait! Let's re-observe the frame and Imagine..."*
*1. A **pit** ahead …*
*2. Detour and get closer*
*3. View at the trunk…*

**"Left, Forward, Camera: [0, 10]"**

Figure A3: **Qualitative example of lookahead and review.** The agent understands the environment (1–2), simulates future states (3), and refines its decision through internal review before acting (4), successfully avoiding a hidden hazard.

| Tokenizer | Compression Ratio | Quantization | MS-COCO | | | | ImageNet-1K | | | FFHQ | | | CelebA-HQ | | |
|---|---|---|---|---|---|---|---|---|---|---|---|---|---|---|---|
| | | | PSNR↑ | SSIM↑ | rFID↓ | PSNR (Minecraft)↑ | PSNR↑ | SSIM↑ | rFID↓ | PSNR↑ | SSIM↑ | rFID↓ | PSNR↑ | SSIM↑ | rFID↓ |
| Open-MAGVIT2 Luo et al. (2024) | 16 × 16 | LFQ | 30.06 | 0.502 | 6.649 | 27.21 | 29.62 | 0.398 | 2.701 | 31.77 | 0.774 | 1.994 | 32.36 | 0.844 | 2.865 |
| LlamaGen Sun et al. (2024) | 8 × 8 | VQ | 30.71 | 0.616 | 4.123 | 28.93 | 30.28 | 0.498 | 1.403 | 33.39 | 0.868 | 0.701 | 34.82 | 0.937 | 0.502 |
| LlamaGen Sun et al. (2024) | 16 × 16 | VQ | 29.93 | 0.491 | 6.077 | 27.06 | 29.81 | 0.448 | 1.657 | 31.58 | 0.772 | 1.366 | 32.18 | 0.837 | 1.113 |
| Cosmos-Tokenizer-DI Agarwal et al. (2025) | 8 × 8 | FSQ | 31.74 | 0.730 | 4.564 | 30.84 | 31.73 | 0.725 | 1.841 | 35.35 | 0.892 | 0.555 | 37.77 | 0.948 | 0.261 |
| Cosmos-Tokenizer-DI Agarwal et al. (2025) | 16 × 16 | FSQ | 30.74 | 0.591 | 12.252 | 29.91 | 30.69 | 0.582 | 6.529 | 33.17 | 0.808 | 7.663 | 33.86 | 0.854 | 5.953 |
| Emu-3 Wang et al. (2024) | 16 × 16 | VQ | - | - | - | 24.16 | - | - | - | - | - | - | - | - | - |

Table A3: **Comparison of Tokenizers across different benchmarks.** PSNR, SSIM, and rFID are measured on MS-COCO, ImageNet-1K, FFHQ, and CelebA-HQ datasets. PSNR for Minecraft images is provided separately.

**Benchmarks and splits.** All embodied evaluations are conducted in the Minecraft simulator following the STEVE-1 protocol (Lifshitz et al., 2023). Unless otherwise specified, **training seeds** and **evaluation seeds** are disjoint (training: *list in Appendix*, evaluation: random seeds per episode). Each reported number is averaged over $N$ evaluation episodes per task (*we set $N = 50$*).

**Settings: Manual vs. Tool.** **Manual** disables iron tools and requires bare-hand interactions; **Tool** enables iron tools to shorten long loops while preserving task logic. We evaluate both settings to cover distinct difficulty regimes.

**Iteration, sample, and efficiency.** One **iteration** denotes a single forward step (one frame) of the policy during evaluation. A **sample** denotes a collected unit in material-gathering tasks (*e.g.*, number of wood blocks). We report **samples per iteration** as data-efficiency (higher is better).

| Method | VPT Baker et al. (2022) | DreamerV3 Hafner et al. (2023) | DECKARD Nottingham et al. (2023) | DEPS Wang et al. (2023c) | Plan4MC Yuan et al. (2023) | Voyager Wang et al. (2023a) | STEVE Lifshitz et al. (2023) | **RIG (Ours)** |
|---|---|---|---|---|---|---|---|---|
| **Demos** | Videos | None | Videos | None | None | None | Videos | Videos |
| **Rewards** | Sparse | Dense | Sparse | None | Dense | None | None | None |
| **Observations** | Pixels Only | Pixels & Meta | Pixels & Inventory | Feedback & Inventory | Pixels & Meta | Feedback & Meta & Inventory | Pixels & Feedback & Meta & Inventory | Pixels Only |
| **Actions** | Keyboard & Mouse | Discrete | Keyboard & Mouse | Keyboard & Mouse | Discrete | Code | Code | Keyboard & Mouse |
| **Reasoning** | | | | ✓ | | ✓ | ✓ | ✓ |
| **Generation** | | | | | | | | ✓ |
| **Extra Database** | | | | | 9 | 172 | 210 | - |

Table A4: **Comparison between RIG (Ours), and existing works.** This system-level comparison of LLM-based and RL-based methods focuses on data sources, reward setup, observation type, action representation, iterative planning, and skill database usage.

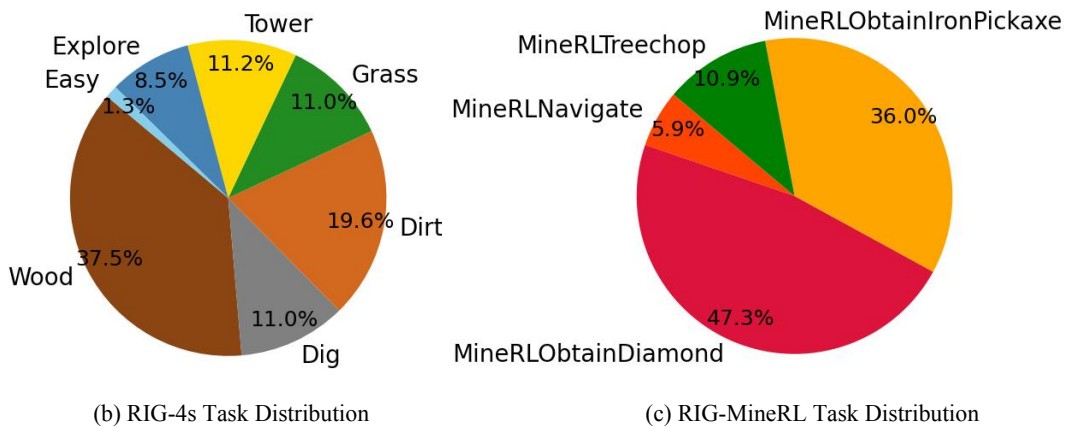

(b) RIG-4s Task Distribution      (c) RIG-MineRL Task Distribution

Figure A4: **Task distribution.** Our datasets include various embodied tasks with varying complexity, ensuring strong generalization across downstream goals.

**Task success metrics.** For **collection** tasks we report the number of collected samples; for **exploration** tasks we report success rate (%). Success is defined as: *Dig*: depth $z \in [7, 20]$; *Explore*: travel distance $\geq 300$; *Tower*: height $\geq 15$.

**Language and reasoning metrics.** **Reasoning** is a blind grade produced by GPT-4o with a legality weight $\{0.5 \text{ (illegal)}, 1 \text{ (legal)}\}$ to penalize unsafe or infeasible plans. **Understanding** is graded via Minecraft QA. **Score-Static** is accuracy on STEVE-21K static QA (0/1, averaged, then rescaled to [0,10]). We release the exact prompts and grading rubric in the Appendix to ensure reproducibility.

**On claims vs. GPT-4o.** When stating that our model "surpasses GPT-4o," we strictly refer to the above *task-specific rubric* and *blind grading* setting, where GPT-4o is *not* evaluating its own responses. We further use alternative prompts and cross-checkers to mitigate grader bias.

## A.4   DATA DISTRIBUTION

Figure A4 visualizes the task distribution across our training datasets, which cover a spectrum of embodied scenarios such as resource collection, tower building, and exploration. As task complexity increases, we progressively expand the dataset size to ensure adequate supervision. Notably, harder tasks like building structures require significantly more data than simpler ones like gathering materials, highlighting the varying difficulty levels and skill composition in our training corpus.

| Type | Model | # LLM Params | Core | | | | VQA | | Exams | |
|------|-------|--------------|------|------|------|------|------|------|------|------|
| | | | POPE ↑ | MME-P ↑ | MMB ↑ | SEED ↑ | VQAv2$_{(test)}$ ↑ | GQA ↑ | MMMU ↑ | MM-Vet ↑ |
| *Und. Only* | | | | | | | | | | |
| *Und. Only* | LLaVA-v1.5-Phi-1.5 Xie et al. (2024) | 1.3B | 84.1 | 1128.0 | - | - | 75.3 | 56.5 | 30.7 | - |
| *Und. Only* | LLaVA Liu et al. (2024c) | 7B | 76.3 | 809.6 | 38.7 | 33.5 | - | - | - | 25.5 |
| *Und. Only* | LLaVA-v1.5 Liu et al. (2024b) | 7B | 85.9 | 1510.7 | 64.3 | 58.6 | 78.5 | 62.0 | 35.4 | 31.1 |
| *Und. Only* | InstructBLIP Dai et al. (2023) | 7B | - | - | 36.0 | 53.4 | - | 49.2 | - | 26.2 |
| *Und. Only* | Emu3-Chat Wang et al. (2024) | 8B | 85.2 | - | 58.5 | 68.2 | 75.1 | 60.3 | 31.6 | - |
| *Und. Only* | InstructBLIP Dai et al. (2023) | 13B | 78.9 | 1212.8 | - | - | - | 49.5 | - | 25.6 |
| *Und. and Gen.* | | | | | | | | | | |
| *Und.&Gen.* | Show-o Xie et al. (2024) | 1.3B | 73.8 | 948.4 | - | - | 59.3 | 48.7 | 25.1 | - |
| *Und.&Gen.* | LWM Liu et al. (2024a) | 7B | 75.2 | - | - | - | 55.8 | 44.8 | - | 9.6 |
| *Und.&Gen.* | VILA-U Wu et al. (2024) | 7B | 85.8 | 1401.8 | - | 59.0 | 79.4 | 60.8 | - | 33.5 |
| *Und.&Gen.* | Janus Chen et al. (2025) | 1.3B | 87.0 | 1338.0 | 69.4 | 63.7 | 77.3 | 59.1 | 30.5 | 34.3 |
| *Und.&Gen.* | **RIG (Ours)**[‡] | 1.3B | 82.8 | 1302.0 | 68.9 | 59.0 | 77.1 | 59.4 | 30.9 | 35.0 |

Table A5: **Comparison with state-of-the-arts on multimodal understanding benchmarks**. "Und." and "Gen." denote "understanding" and "generation", respectively.

## A.5 Component Comparison

As summarized in Table A2 and Table A4, we compare RIG to prior works along multiple dimensions, including input modality, action granularity, and reasoning capabilities. Unlike prior methods relying on handcrafted API actions or curated codebooks, RIG operates solely on raw pixels and outputs keyboard-mouse controls, offering higher flexibility and lower task bias. Notably, our design unifies reasoning and generation into a single transformer policy with self-review and imagination steps, offering better trajectory-level coherence and enabling multi-turn lookahead.

## A.6 Tokenizer and Base Model Selection

We adopt LlamaGen $16 \times 16$ VQ tokenizer and Janus-1.4B as our vision and language backbone. Table A3 reports their favorable reconstruction quality (PSNR 27.06) and semantic alignment. Janus uses a dual loss combining RGB and SigLIP-guided feature reconstruction, while LlamaGen provides discrete, compression-friendly tokens. Together, they form a scalable pipeline for visual imagination and reasoning, trained with simple cross-entropy objectives.

## A.7 General VQA Benchmark Results

**Setup.** To assess whether embodied specialization compromises general multimodal ability, we evaluate RIG (1.3B) on a standard VQA/exam suite covering **Core** perception (POPE, MME-P, MMB, SEED), **VQA** (VQAv2, GQA), and **Exams** (MMMU, MM-Vet). Thanks to Janus Chen et al. (2025), all results and settings about the compared baselines are coordinated with it. And we compare with parameter-matched against Janus-1.3B for fairness, see Table A5.

**Overall parity with slight gains on exam-style reasoning.** Across **VQA** benchmarks, RIG remains on par with Janus (VQAv2: 77.1 vs. 77.3; GQA: **59.4** vs. 59.1), indicating *no catastrophic forgetting* of generic visual QA. On **Exams** that emphasize compositional reasoning and multi-hop understanding, RIG shows small but consistent improvements (MMMU: **30.9** vs. 30.5; MM-Vet: **35.0** vs. 34.3). We attribute these gains to the explicit reasoning curriculum (S2) and lookahead reviewing (S3–S4), which strengthen structured CoT and long-horizon inference even outside the embodied environment.

**Core perception remains competitive.** On **Core** perception metrics, RIG is competitive with Janus (MMB: 68.9 vs. 69.4). Slight drops on POPE and MME-P (82.8 and 1302.0) are within typical variance for domain-specialized finetuning and can be further mitigated by a short generic-VQA refresh stage or lightweight anti-hallucination regularization (left for future work). Importantly, these changes do not translate into regressions on downstream VQA/Exam tasks.

**Takeaways.** (1) Embodied adaptation *does not* erode general VQA competence, (2) reasoning-centric training provides transferable benefits on exam-style evaluations, (3) any minor perception drift is small and local, while end-task metrics remain stable or improved. Together, Table A5 supports our claim that RIG maintains broad multimodal understanding while gaining domain-relevant reasoning skills.

A.8    QUALITATIVE RESULTS AND CASE STUDY

Figure A3 demonstrates the full inference cycle of RIG, where the agent understands the scene, reasons about its next move, simulates imagined outcomes, and conducts self-review before taking real action. In this wood-chopping task, the agent first identifies a tree in front, then reasons that moving forward seems viable. However, by simulating future states, it spots a hidden pit and triggers a self-correction: "`Wait! Let's re-observe...`". It updates its decision to `Left, Forward, Camera: [0, 10]: right`, successfully avoiding the hazard. This highlights our agent's ability to perform proactive planning, visual forecasting, and risk-aware correction through imagination and reviewing.

**Comparison with GPT-4o image generation updated version.**   Figure A5 further compares RIG-*lookahead* with GPT-4o image generation updated version. Both receive similar prompt and visual input. While GPT-4o generates a visually plausible prediction, it incorrectly judges the distance to the tree, prematurely issuing an `attack` command that leads to a deadlock. It continues to hallucinate progress without correcting the faulty assumption. In contrast, RIG accurately detects that the tree is blocked and unreachable, reasons about terrain features, and adjusts its position before action. The generated image aligns with the actual environment response, showing stronger spatial consistency and robustness in long-horizon decision-making.

**Data pipeline comparison between GPT-4o and Qwen3-VL-8B-Instruct.**    To assess whether a lighter open-source VLM can replace GPT-4o in our data pipeline, we run a small-scale comparison using Qwen3-VL-8B-Instruct Yang et al. (2025) as a drop-in annotator under the *same* prompts and first-person observations. Figure A6 shows two representative cases from the "collect more wood" task, where the next action is "jump" (top) or "forward" (bottom). In both examples, GPT-4o (right) produces concise, first-person reasoning that is tightly grounded in the visual input, correctly describing the relative position to the target tree and why the proposed action moves the agent toward a successful state. By contrast, Qwen3-VL-8B-Instruct (left) often adopts a third-person narrative ("the player"), hallucinates objects or properties (e.g., claiming the agent is holding a wooden pickaxe when the hands are empty), and generates long, redundant text that only loosely matches the current frame and task. Colored highlights mark correct statements (green), factual errors (red), and redundant or off-task content (gray). These qualitative results indicate that Qwen3-VL-8B-Instruct currently produces substantially lower-quality reasoning labels in our setting, which is why we rely on GPT-4o for the prototype. We believe that future stronger open-source VLMs, or student models distilled from GPT-4o, may close this gap and make the pipeline more accessible to the broader community.

A.9    MULTI-MODAL UNDERSTANDING EVALUATION

We further evaluate RIG on the STEVE-21K (Zheng et al., 2023) benchmark, testing its general world knowledge and Minecraft-specific understanding. Drawing from the Minecraft Wiki and Reddit corpus, the dataset spans multiple knowledge dimensions:

- **World Understanding:** Terrain, entities, and biome behaviors.
- **Player Mechanics:** Combat, mobility, and health systems.
- **Survival Strategies:** Food sourcing, shelter, and threat avoidance.
- **Resource Management:** Gathering, mining, and inventory use.
- **Crafting and Construction:** Recipes and structural planning.
- **Tool Usage:** Equipment selection and upgrades.

We evaluate with 1000 QA pairs, categorized as: World & Entities (332), Mechanics & Survival (152), Knowledge & Discovery (108), Crafting (219), Tools (169), and Miscellaneous (20). Our model demonstrates strong accuracy and reasoning coherence across categories.

A.10    MULTI-TURN VISUAL REASONING FORMAT

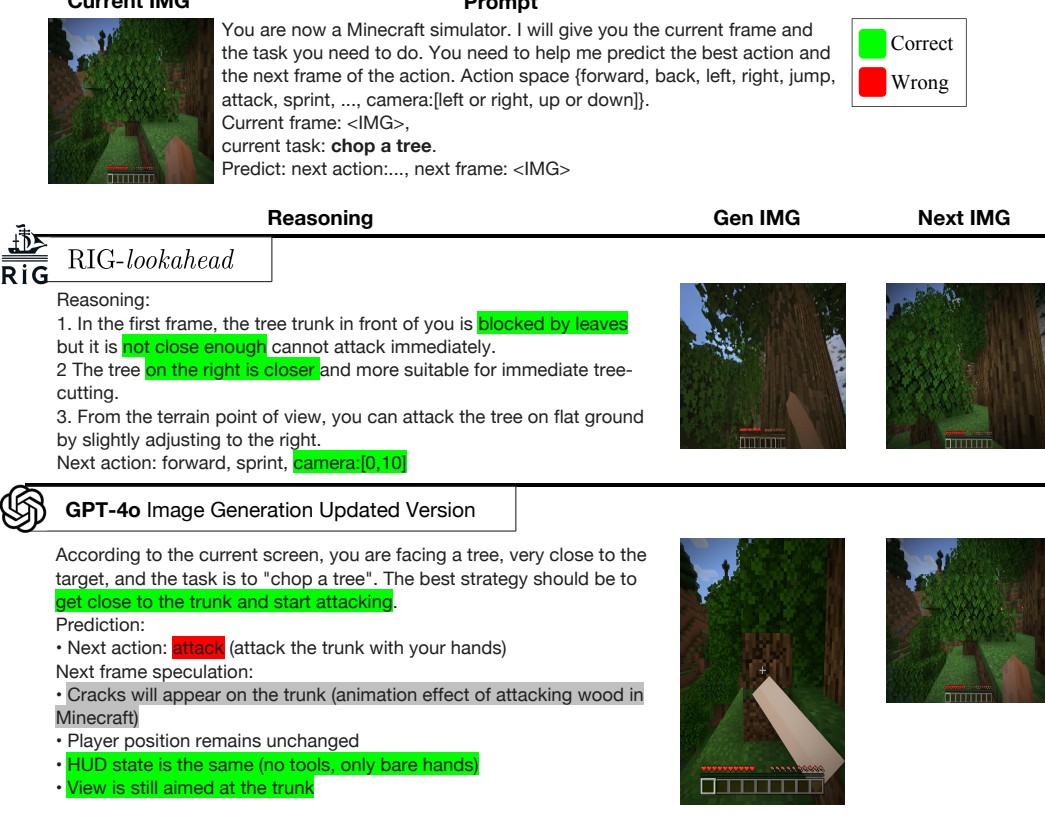

Figure A5: **Case study on reasoning and imagination.** Given the same input and prompt (`chop a tree`), RIG reasons and imagines future states to choose a reachable tree and adjust position before acting. GPT-4o, despite high visual quality, misjudges the distance, executes an invalid action, and fails to revise its plan.

To supervise step-level visual reasoning, we define a structured multi-turn dialogue format, as shown in below. Each entry logs the task instruction, prior action, current frame, reasoning, next action, and imagined future frame. This design aligns with autoregressive generation and supports fine-grained analysis and supervision.

- **Task Instruction:** Natural language goal (*e.g.*, "build a tower").
- **Previous Action:** Last executed action (*e.g.*, "camera:[0,10]").
- **Current Frame:** Visual observation from the environment.
- **Step Reasoning:** Textual reasoning for the next decision.
- **Next Action:** Predicted action.
- **Next Frame:** Imagined visual result of the action.

## A.11 ENVIRONMENT DETAILS

We use Minecraft as the testbed for embodied agents due to its open-ended nature and support for low-level human-like interactions. Agents act through egocentric RGB images and execute actions using keyboard and mouse inputs, making the environment ideal for sequential decision-making.

Our experiments are based on MineRL (Guss et al., 2019) v1.0 (Minecraft 1.16.5), which provides agents with first-person RGB observations and removes access to any privileged information. This

version aligns with setups in prior works such as VPT (Baker et al., 2022) and STEVE-1. Agents only perceive visual inputs and interact through low-level actions, resembling human play

In our experiments, we use S = 64 environment seeds per task, covering 6 different biomes (plains, forest, taiga, mountains, desert, and snowy variants). After filtering, no single seed contributes more than K = 5 trajectories. The filtering is done in three steps: 1. Task success: We discard all trajectories that do not satisfy the task success criterion. And then **rollout 6 successful** trajectories on each node where failure happens. 2. Seed diversity: We enforce the per-seed cap (K = 5) to avoid over-representing any specific layout or biome. 3. Reasoning quality: For a stratified subset of trajectories, we manually check that the textual reasoning is consistent with the visual scene and the executed actions; trajectories with hallucinated or clearly inconsistent reasoning are removed.

### A.12   Observation and Action Space

The agent receives $640 \times 360$ RGB images rendered from a first-person view with a 70-degree field of view. When the inventory is opened, the GUI and mouse cursor are visible. No voxel, depth, or structured APIs (*e.g.*, "craft", "smelt") are used.

As shown in Table A4, the action space includes 14 types of keyboard and mouse operations, covering movement (WASD), item use, inventory management, and camera rotation (yaw, pitch). These mirror human inputs and do not rely on high-level abstractions.

For the `camera` action, which is originally a 2D continuous vector $[a, b]$ representing pitch and yaw in $[-180°, 180°]$, we quantize it into discrete steps of 5 degrees to adapt to language models, where $a{<}0/a{>}0$ denote looking up/down and $b{<}0/b{>}0$ denote turning left/right; all other actions are binary (0/1), indicating whether to execute them.

### A.13   Environment Settings and Rules

To ensure diversity and generalization, each episode is initialized with a random agent position and world seed, exposing the model to varying terrains, structures, and objectives.

We adapt the MineRL (Guss et al., 2019) environment to support instruction-based initialization via a custom `chat` action, enabling programmatic task injection and faster evaluation cycles.

**Programmatic Evaluation.** For instruction-following tasks, we fix the seed and biome across different models to ensure a fair comparison. Each experiment runs under conditions most favorable for the given instruction, and multiple seeds are used across different trials.

**Long-Horizon Tasks with Command Switching.** In these scenarios, agents operate in a shared environment initialized with a common seed and biome. To reduce noise and stabilize training/testing, we apply the following environment rules:

- `/difficulty peaceful`: Disables hostile mobs.
- `/gamerule doDaylightCycle false`: Locks the environment in perpetual daytime.
- `/gamerule keepInventory true`: Prevents item loss upon death.

These configurations help maintain consistency across runs while preserving the task's difficulty and realism. They also support efficient benchmarking of embodied reasoning and planning in long-horizon, open-ended environments.

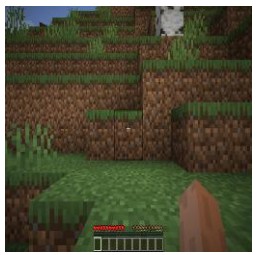

**Prompt**

<System> You are an experienced Minecraft player with the first-person view. You will get the task and the correct next action for the task, and you should make proper reasoning for the next action.

Your task is collecting more wood, the next action is "jump". Please make the reasoning on the first-person view for the next action to realize the task.

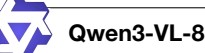 **Qwen3-VL-8B-Instruct**

The player is currently holding a wooden pickaxe and is standing on a grassy ledge. To get wood, they need to access more trees or wooden blocks, which are located higher up on the terrain. Jumping will allow them to reach the elevated ground where more wood is available, enabling them to gather more resources and progress toward their task. Therefore, the next logical action is to jump to access higher ground and continue collecting wood.

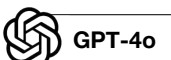 **GPT-4o**

I should jump to climb onto the dirt block directly in front of me. The birch tree, which I can use to get wood, is on the higher ground ahead. By jumping up this step, I move closer to the tree so that I can reach its trunk and start punching it to collect wood.

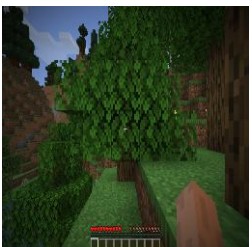

**Prompt**

<System> You are an experienced Minecraft player with the first-person view. You will get the task and the correct next action for the task, and you should make proper reasoning for the next action.

Your task is collecting more wood, the next action is "**forward**". Please make the reasoning for the next action to realize the task.

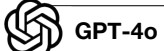 **Qwen3-VL-8B-Instruct**

The player is currently holding a wooden pickaxe and is standing near a tree — a key source of wood in Minecraft. However, to efficiently gather wood, they need to position themselves directly in front of the tree to break its trunk or branches. The "forward" action moves the player closer to the tree, allowing them to:
Better aim their pickaxe at the tree trunk or leaves.
- Access the tree from a more optimal angle for harvesting.
- Prepare to break the tree and collect wood planks or logs.
Since the task is to get wood, moving forward is the logical next step to reach the tree and begin harvesting it. This action directly contributes to completing the task by positioning the player for interaction with the wood source.

**GPT-4o**

I should move forward to get closer to the tree right in front of me. From this distance I still can't reach the trunk blocks to punch them. Walking forward puts me directly next to the tree, so on the next steps I can start breaking the wood blocks and collect logs to complete the "getting wood" task.

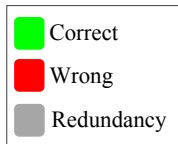

| | |
|---|---|
| 🟩 | Correct |
| 🟥 | Wrong |
| ⬜ | Redundancy |

Figure A6: **Qualitative comparison of reasoning labels produced by Qwen3-VL-8B-Instruct (left) and GPT-4o (right).** Given the same first-person observation and prompt (top: next action is "jump"; bottom: next action is "forward") in a wood-collecting task, GPT-4o generates grounded first-person reasoning that correctly describes the relative position to the target tree and the effect of the action. Qwen3-VL-8B-Instruct, however, often adopts a third-person perspective, hallucinates objects (e.g., a wooden pickaxe), and produces redundant or off-task text. Green, red, and gray highlights denote correct, wrong, and redundant statements, respectively.

---

**Prompt Format for Minecraft Conversation Reasoning**

**SYSTEM:**

You are assisting in generating step-by-step reasoning data for Minecraft embodied agents in complex tasks like **"build a tower"**. Each step should contain (1) the reasoning process, (2) the predicted next action, and (3) the generated next image frame. We follow a multi-turn conversation structure with visual context.

**Input Format:**

- **Task**: Given current task like "build a tower"
- **Previous Action**: e.g., camera:[0,10]
- **Current Frame**: Image observation

**Human Prompt (Input):**

```
<image>
Please make reasoning of action and predict the next action
based on the current frame, current task and previous action,
and generate the next frame based on the current frame and the
next action.
The current task: build a tower.
The previous action: camera:[0,10].
```

**RIG Output (Response):**

> **The reasoning:** In the current frame, the task is to build a tower, but no suitable material is directly targeted. The player may be exploring for logs or stone. The previous action moved the camera upward to scout surroundings...
> **Next action:** left, sprint, and next frame: <image>

**Special Cases: Lookahead and Imagination**

- When the model performs internal reflection or imagines the future, use the special token: `<Imagine:>` to prefix the reasoning block.
- Example: `<Imagine:>` Reasoning forward $n$ steps into future environment outcomes.

**Fields to be included in JSON:**

- name, id, action, images, conversations, subtask (optional)

**Example JSON Structure:**

```json
{
  "name": "build_a_tower_seed203",
  "id": 10,
  "action": "left, sprint",
  "conversations": [
    {
      "from": "human",
      "value": "<image>\n <Imagine:> Please make reasoning of
      action...
      task: build a tower...",
      "images": ["..._10_current.png"]
    },
    {
      "from": "RIG",
      "value": "The reasoning: ... Next action: left, sprint,
      sand next frame: <image>",
      "images": ["..._10_next.png"]
    }
  ]
}
```

