# OpenReview forum: "RIG: Synergizing Reasoning and Imagination in End-to-End Generalist Policy"
_ICLR.cc/2026/Conference — ICLR 2026 Poster_

### Official Review · Reviewer_tgFQ · 2025-10-29

**Soundness:** 3
**Presentation:** 3
**Contribution:** 3
**Rating:** 6
**Confidence:** 4

**Summary:**

This paper proposes RIG (Reasoning and Imagination in Generalist policy) , an end-to-end Generalist policy framework, for the first time, text reasoning, low-level action control, and visual imagination are jointly modeled in the same Transformer architecture. The authors use GPT-4o to inject reasoning and reflective annotations into trajectories through a progressive data construction process (S 0-s 4) , and train two versions: the reasoning-only RIG-basic and the RIG-lookahead with the ability of“Forward-reflection”. In the Minecraft environment, RIG achieved much better sample efficiency (17 × boost) and task success rates than existing methods such as VPT, Steve-1, MineDreamer, with only 111 h of training data, and the performance of the proposed method was better than existing methods, test-time scaling is also supported to enhance decision robustness.

**Strengths:**

- Outstanding originality: unifying “Reasoning” and “World Model Imagination” into a single autoregressive Transformer rather than splicing multiple specialized modules (e.g. , VLM + VGM) is a significant innovation in architectural design. In particular, the“Dream-review” mechanism generates a correction trajectory through GPT-4o and skillfully converts failure experiences into training signals, which is a novel paradigm of data enhancement and self-monitoring.
- The proposed four-stage data construction process (S0-S4) is logically clear and reproducible, and each stage has a clear goal (from basic alignment → Inference Injection → failure reflection → temporal alignment). The experimental design was comprehensive, covering ablation studies, scalability analysis (data volume, number of iterations, number of prospective steps), multi-task evaluation (collection/exploration), quality of generation (FID/PSNR), and reasoning ability (VQA), and the results were analyzed, providing strong support.
- Significant practical implications and potential: exceeding the baseline of relying on 2000 hours of video with only 111 hours of data, demonstrating the tremendous improvement in sample efficiency of “Collaborative modeling, inference, and imagination”; it has important reference value for embodied intelligent scenarios with scarce data. Lookahead also provides a viable path to reducing real-world trial-and-error.
- Clear problem localization and comparison: the paper accurately points out the shortcomings of the existing works (e.g., Voyager, MineDreamer) in“End-to-end optimization” and“Reasoning-imagination collaboration”, and provides a new perspective for future research, and through the intuitive comparison of four types of agent architectures in Figure 1, the contribution positioning is clear.

**Weaknesses:**

- All experiments were conducted in Minecraft, with clear rules and a highly structured (boxed) visual style, with gaps compared to real-world or more complex simulators (e.g., Habitat, Isaac Gym). The authors do not discuss the applicability of the method to continuous action spaces, real images, or dynamic physical environments, which weakens the universality of the conclusion.
- S2(inferential tagging) and S3(reflective tagging) rely heavily on GPT-4o to artificially generate trajectories, but do not account for tagging costs (e.g., token consumption, manual screening time), and do not account for tagging costs (e.g, nor has there been any attempt to replace it with a more lightweight model, such as an open source LLM. This may affect the scalability and practicality of the method, especially for researchers with limited resources.

**Questions:**

- Is the RIG's architecture suitable for non-boxed, continuous-control, or real-world robotic environments? Can you provide preliminary migration experiments (e. g. Carla, MetaWorld) in the appendix or in future work?
- Have you tried annotating trajectories with smaller open-source models, such as LLAMA-3-8B with visual adapters? How does the quality of its build affect final performance? This is critical for community replication.
- The current “Imagination” is only used to generate modified inferences, not to directly optimize action strategies (such as policy gradients based on imaginary trajectories). Are there plans to integrate RIG with reinforcement learning for tighter imagination-based planning?

---

> ### Author Response · Authors · 2025-11-20
> **Comments on Weaknesses**
>
> We sincerely thank the reviewer for the thoughtful comments on generalization beyond Minecraft, annotation cost and scalability, and the possibility of integrating RIG with reinforcement learning. We address each point below.
>
> ### **A. (W1, Q1) Applicability beyond Minecraft / continuous-control / real-world settings**
>
> We fully agree that evaluating embodied agents beyond a single, voxelized environment is important, and we appreciate the examples of Habitat, Isaac Gym, CARLA, and MetaWorld.
>
> **Architecture-wise, RIG is not tied to Minecraft.** The core components of RIG: (1) textual reasoning, (2) visual imagination via future-state prediction, and (3) action selection guided by internal review, are environment-agnostic. In our implementation:
>
> - The **visual encoder / decoder** can operate on arbitrary first-person images or rendered observations, not only Minecraft frames.
> - The **action head** is a small task-specific module; in Minecraft it predicts discrete actions, but in principle it can be replaced by a continuous-control head (e.g., predicting joint torques or continuous steering commands).
> - The **lookahead and review mechanism** only assumes that imagined states and actions can be scored by success/failure–style signals, which remain meaningful in continuous and real-world settings.
>
> In this paper, we chose Minecraft as a **prototype testbed** because it provides:
>
> (1) controllable environment seeds and biomes,
>
> (2) long-horizon tasks with clear success conditions, and
>
> (3) a standardized interface that allowed us to run extensive experiments within our resource budget.
>
> Due to the limited submission timeline and compute, we were not able to complete a sufficiently rigorous set of experiments on additional simulators (e.g., CARLA/MetaWorld) to confidently include them in the main paper. We definitely agree that **demonstrating RIG on non-boxed, continuous-control simulators and real robot images is an important direction** for future work, and we are actively working toward such extensions.
>
> ### **B. (W2, Q2) Annotation cost and replacing GPT-4o with lighter open-source models**
>
> We appreciate the reviewer’s concern about the practicality and reproducibility of using GPT-4o for S2 (inferential tagging) and S3 (reflective tagging).
> Annotation cost and scalability.
>
> For S2 and S3, GPT-4o is used offline to annotate reasoning traces and reviews based on first-person observations and trajectories. This cost is:
>
> - **Only used** over the data pipeline for the training process.
> - Linear in the number of annotated steps. In the revision, we have added a short paragraph in the appendix reporting our total annotation volume (number of prompts and tokens) and relating it to the training compute (we already report ~704 GPU hours for RIG-basic and ~280 GPU hours for RIG-lookahead on 64×A100 80GB).
>
> We agree that, for groups with more limited resources, it is crucial to understand how far one can go with **lighter, open-source annotators**.
> Preliminary attempt with a smaller open-source VLM.
> Motivated by this concern, we conducted an initial experiment using Qwen3-VL-8B-Instruct as a drop-in replacement for GPT-4o in our data pipeline (same prompts, same first-person observations). We summarize our findings (detailed in Appendix A.8 and Figure A.7 in the revised version):
>
> - The model often failed to properly adopt the first-person vantage point, leading to reasoning that confused the agent with an external observer.
> - We observed frequent factual errors about the visual scene, such as describing “empty hands” when the avatar is clearly holding a wooden pickaxe, or hallucinating objects that are not present.
> - The generated reasoning was much noisier and off-task, with long, messy text that did not align well with the underlying actions and goals.
>
> Because these issues produced substantially lower-quality reasoning labels, we decided not to use these annotations for training RIG in the current paper. At the same time, we view these experiments as encouraging: they suggest that stronger open-source VLMs or lightweight student models distilled from GPT-4o might become viable annotators in the near future.

---

> ### Author Response · Authors · 2025-11-20
> **Comment on Question**
>
> ### **(Q3) Using imagination for policy optimization / reinforcement learning**
> We very much agree that combining RIG with reinforcement learning and policy-gradient–style updates based on imagined trajectories is a promising extension.
> In the current submission:
>
> -  We primarily use supervised learning and **reject-sampling fine-tuning (RFT)** on real and imagined trajectories to obtain RIG-lookahead. This can be viewed as a simplified, single-step, offline RL-style procedure, where imagined trajectories are labeled and filtered, and the policy is updated to prefer higher-quality futures.
> - The **imagination module** is used to generate **modified inferences and internal reviews**, which guide action selection at inference time, but we do not yet backpropagate a full RL objective through multi-step imagined rollouts.
>
> We see two natural extensions:
>
> 1. **On-policy RL with imagination-based planning.** Starting from a strong SFT/RFT checkpoint (such as RIG-lookahead), one could use the imagination and review heads to estimate returns or advantages for multiple candidate plans and then apply policy-gradient or actor–critic updates. This would more tightly couple imagination with optimization, at the cost of additional compute.
> 2. **Model-based RL with learned world model.** The visual prediction head in RIG already functions as a world model in latent space. Using it for imagined rollouts and applying model-based RL (e.g., value iteration or trajectory optimization in imagination) is a natural next step.
>
> Due to resource constraints and the focus on establishing a solid prototype in one environment, we did not include such RL experiments in the current paper. We will add a short discussion section stating that **RIG is compatible with model-based and imagination-based RL**, and that integrating these directions is a key line of future work.

---

> > ### Comment · Reviewer_tgFQ · 2025-11-22
> >
> > Thanks to the author's answers to my questions, almost all my doubts were resolved and I maintained my original positive score

---

> > > ### Author Response · Authors · 2025-11-22
> > >
> > > We sincerely thank the reviewer for their careful reading of our work and are grateful that our clarifications have addressed the main concerns while preserving the positive evaluation.

---

### Official Review · Reviewer_JjLn · 2025-10-31

**Soundness:** 2
**Presentation:** 2
**Contribution:** 3
**Rating:** 4
**Confidence:** 3

**Summary:**

This paper introduces RIG, a generalist policy that aims to synergize Reasoning (text), Imagination (future frames), and action into a single, end-to-end autoregressive model. It is trained using a progressive data pipeline that distills reasoning and review labels from large language models (GPT-4o) onto existing agent trajectories. During inference, RIG can "dream" potential outcomes, "review" these imagined futures, and "self-correct" its action plan, enabling a form of test-time planning. The authors claim this synergy achieves SOTA performance and a 17x sample efficiency improvement on Minecraft tasks .

**Strengths:**

1. The paper's core idea of a single, end-to-end model that unifies reasoning, imagination, and action in one autoregressive sequence is a compelling vision for generalist agents.
2. Table 2 provides good ablation evidence that the joint training is synergistic, where adding Reason capabilities improves Generation quality, and adding Gen capabilities improves Reasoning scores.
3. The "dream-review" mechanism is a novel and interesting contribution. It provides a concrete data-driven method for teaching the model to perform internal simulation, spot its own potential failures, and then self-correct its plan before execution.

**Weaknesses:**

1. The paper's single most prominent claim is its "17x sample efficiency" (111h of data) compared to baselines like STEVE-1 (2000h). However, the paper's own data pipeline explicitly requires a "pretrained policy, STEVE-1" to generate the S1 (Vision-Action) dataset. Furthermore, the S3 stage requires a "superior-performing policy STEVE-1" to generate positive trajectories for its corrective "dream-review" data. Therefore, RIG's 111h data budget requires a pre-existing 2000h STEVE-1 model as a prerequisite. The 2000h data cost of this "teacher" model is hidden and not included in the 111h total.
2. The paper frames RIG as a novel learning agent that "synergizes reasoning and imagination". However, the data pipeline shows that the "reasoning" (S2) and "reviewing" (S3) are not learned from environmental interaction but are distilled from GPT-4o, and the vision-action training data are collected through a pretrained policy (STEVE-1) in S1 and S3. The agent is learning to imitate GPT-4o's reasoning and STEVE-1's actions. This is more of a distillation strategy rather than a novel, sample-efficient learning paradigm that the paper claims to be.

**Questions:**

The imagination and self-review pipeline during inference is not that clear, can you elaborate it more concretely?

---

> ### Author Response · Authors · 2025-11-20
> **Comments on Weaknesses**
>
> We thank the reviewer for carefully raising these important conceptual questions about sample efficiency, the role of STEVE-1 and GPT-4o, and the inference-time pipeline. We clarify our standpoint and revise the framing accordingly.
>
> ### **A. (W1) On the 17× sample efficiency and the role of STEVE-1**
>
> We agree that it is important to be explicit about what is counted in the “17× sample efficiency” claim.
> - In the current paper, **“111h vs. 2000h”** is defined strictly in terms of environment interaction used to train the new agent, following the **common convention** in teacher–student and learning-from-demonstrations settings [1] [2] [3]: a teacher (or expert policy) is assumed given, and sample efficiency is measured with respect to the **learner’s own training budget**. We will explicitly mention this convention and add references to knowledge distillation and learning-from-demonstrations works where teacher training cost is not folded into the student’s data budget.
> - Concretely, RIG and STEVE-1 are both trained approximately “from scratch” as an embodied agent in Minecraft, **without superior prior knowledge** specific to this environment. STEVE-1’s 2000h interaction produced by human is already part of the baseline we compare to. When we report that RIG achieves comparable or better performance with 111h of interaction, we are comparing the additional environment interaction needed by the new agent to reach the reported performance.
> - Importantly, the teacher we use for data collection (STEVE-1) is the same model as our baseline. Thus, any accounting scheme that attributes the 2000h training cost to RIG would, by symmetry, also need to attribute it to the STEVE-1 baseline itself. Under this lens, our contribution is that given an already-trained STEVE-1, RIG achieves its improvements with 111h of new interaction, whereas further fine-tuning STEVE-1 alone with the same budget does not match RIG’s performance.
>
> We will revise the wording in the main text to:
>
> (1) explicitly state that the 17× factor refers to learner-side environment interaction,
>
> (2) acknowledge the existence of the teacher’s training cost, and
>
> (3) discuss this assumption as a limitation and a standard practice in distillation-style setups.
>
> ### **B. (W2) Distillation vs. “novel learning paradigm” and the role of GPT-4o / STEVE-1**
>
> Our intention is not to deny that RIG uses data pipelines which can be somehow viewed as distillation, but to clarify what is new about how we structure and use it.
>
> - Sources of supervision. RIG indeed draws from two external sources:
>
> (a) STEVE-1 provides low-level vision–action traces and basic skills;
>
> (b) GPT-4o provides reasoning and review annotations.
>
> However, neither of these alone can solve our long-horizon, reasoning-heavy tasks: STEVE-1 fails on many challenging tasks despite 2000h of interaction, and GPT-4o, while strong at language and vision, is not an executable policy in the Minecraft environment. Figure 5 highlights that neither STEVE-1 nor GPT-4o alone achieves the performance of the final RIG agent.
>
> - Beyond direct imitation. RIG is not simply copying STEVE-1’s actions or GPT-4o’s text. Instead, the three stages (S1–S3) and the imagination–lookahead pipeline use environment feedback and automatic filtering to:
>
>   - extract successful sub-trajectories from STEVE-1,
>   - use GPT-4o’s reasoning only where it is consistent with visual outcomes, and
>   - relabeled imagined trajectories (via success/failure signals and reviewer feedback) to refine the policy. This is closer to a structured distillation + self-correction framework, where the agent learns to jointly reason, imagine, and act in a way that neither teacher can do individually.
> Architectural novelty.
>
> The core contribution we intend to highlight is that RIG **integrates three abilities into a single, deployable model**:
>
> (1) **textual reasoning**,
>
> (2) **visual imagination (future frame prediction)**
>
> (3) **action selection**, all trained under a unified objective and used jointly during inference. This synergy, where imagined futures and internal lookahead of “self-review” guide action choices, is not present in STEVE-1 or GPT-4o, and is responsible for RIG’s improvement on long-horizon tasks.
>
> To avoid overstating the paradigm shift, we will revise the text to more explicitly describe RIG as a sample-efficient, teacher-aided framework that leverages distillation from STEVE-1 and GPT-4o, while emphasizing that the key novelty lies in the reasoning–action-imagination coupling and the lookahead (based on dream-review pipeline), rather than in discarding distillation altogether.
>
> [1] Hinton et al., Distilling the Knowledge in a Neural Network, 2015.
>
> [2] Li et al., Few Sample Knowledge Distillation for Efficient Network Compression, CVPR 2020.
>
> [3] Scheller et al., Sample Efficient Reinforcement Learning through Learning from Demonstrations, PMLR 2020.

---

> ### Author Response · Authors · 2025-11-20
> **Comment on Question**
>
> ### **(Q1) Imagination and self-review pipeline at inference**
>
> Thank you for asking for a clearer description of the inference-time pipeline. We clarify below (and will incorporate a concise version in the main text, with pointers to the appendix figure).
>
> At inference time, no external GPT-4o or STEVE-1 is used. A single RIG model performs:
>
> 1. **Observation and reasoning.** Given the current state (first-person frame + context), the model produces a reasoning trace r_t describing its understanding of the situation and candidate plan.
>
> 2. **Imagination (visual prediction).** Conditioned on the current state and the candidate action sequence(s), the visual prediction head generates imagined future frames for the next K steps (a short rollout), forming imagined trajectories $\tau^{\text{imag}} = (s_t, a_t, s_{t+1}^{\text{imag}}, \dots)$.
>
> 3. **Self-review.** The same model includes a “review” head trained to output review-style feedback and a scalar score for each imagined trajectory, in the format of “this plan will likely succeed/fail because …”. This head was trained using GPT-4o review annotations and environment-derived labels (success, failure, rule-based signals), so at inference it can score trajectories without calling GPT-4o.
>
> 4. **Action selection.** Among the imagined rollouts, RIG selects the trajectory with the highest internal review score and executes only the first action from that trajectory in the real environment.
>
> 5. **Iterate.** The environment advances by one step, generating a new observation, and the process repeats.
> This procedure can be seen as a learned lookahead planner: the model uses its own imagination and internal critic (self-review) to evaluate multiple possible futures before committing to an action. We have clarified this step-by-step process in the main text and refer to Figure A6 for a visual illustration.

---

> ### Author Response · Authors · 2025-11-27
>
> We would like to kindly ask whether our rebuttal has addressed your main concerns or if there are any remaining issues that we could further clarify. If our responses have satisfactorily resolved your doubts, we would be sincerely grateful if you could consider updating your score accordingly. Thank you very much for your time and effort in reviewing our work.

---

### Official Review · Reviewer_bvMP · 2025-11-01

**Soundness:** 3
**Presentation:** 2
**Contribution:** 3
**Rating:** 6
**Confidence:** 3

**Summary:**

The paper proposed RIG, Reasoning and Imagination in an end-to-end Generalist policy method to jointly learn next-n-step imagination, text reasoning, and action predictions in the MineCraft environment. The method leverages the benefit of VLM styled reasoning as well as world model styled imagination aiming to combine all modalities together for a more efficient generalist embodied agent performance. More specifically, the paper proposed a progressive data collection strategy, training a RIG-base vision-action-reasoning model without imagination through a relabeled human trajectory dataset, and a pre-trained policy to collect image-action pairs. It leverages GPT-4o to annotate corresponding reasoning for training. Next, the paper paired negative trajectory and positive trajectory, adopting GPT-4o to generate review & revise annotation and aligned the frames temporally for long horizon training stability. The paper trained the RIG-lookahead model with this data and through rejection sampling fine-tuning. The paper conducted experiments on 6 MineCraft tasks, evaluating the data efficiency, scalability, and performance on embodied, generation, and VAQ reasoning tasks against multiple baselines respectively. The paper shows that the proposed RIG methods significantly improves data efficiency, achieves higher task performance, and is scalable.

**Strengths:**

- The paper is well motivated for embodied tasks through imagination (next frame prediction) before action prediction
- The proposed method jointly trains text reasoning, visual prediction, and action prediction to explicitly model the imagination reasoning before action execution
- The paper proposed a progressive data collection pipeline to enable the training of imagination and reasoning
- The paper conducted thorough experiments to demonstrate the performance, efficiency, and scalability through multiple tasks and comparing against multiple baseline models.

**Weaknesses:**

Questions below.
Suggestions:
1. Ln 207: "All these trajectories are rigorously filtered based on task success, diversity across environment seeds, and manual validation of reasoning quality." -- (minor) might be helpful to explain how 'rigorous', how 'diverse' these procedures are before claiming them.
2. Ln 262 Eq (4): the notation is a little confusing: ""wait! Let's re-observe..." is that part of Y- or Y+?
3. In the Experiment section, the paper highlighted the importance of 'Number of Samples' through multiple figures and tables to highlight the data efficiency. It would be helpful to connect the dots and explain more what is 'number of samples' and and why a higher number indicates better data efficiency
4. The appendix contains a large amount of details, figures, and results. It would be more helpful to organize and highlight the important and relevant material.
5. Ln 1133: "Our model demonstrates strong accuracy and reasoning coherence across categories" -- what are the evidence to back up this claim?
6. Ln 1170: space
7. Ln 1190: VPT (?)
8. Ln 377: As shown in Figure 4? Figure 5?

**Questions:**

1. How generalizable is the method after training when applied on a different task or a different environment than MineCraft?
2. In cases of multiple good answers at certain time point t, does the data collection pipeline only collects a single GT path for training? Does the model only know how to image one path forward?
3. Figure 5 and Section 3.3 S3: when STEVE-1 only has acc 47.4% and RIG-basic has acc 93.4%, how does S3 collect sufficient positive trajectory from STEVE-1 while negative trajectories from RIG-base?

---

> ### Author Response · Authors · 2025-11-20
> **Comments on Weaknesses**
>
> ### **A. (W1) “Rigorously filtered” and “diverse” trajectories**
>
> We appreciate the suggestion to be more concrete here.
>
> In our experiments, we use S = 64 environment seeds per task, covering 6 different biomes (plains, forest, taiga, mountains, desert, and snowy variants). After filtering, no single seed contributes more than K = 5 trajectories.
>
> The filtering is done in three steps:
>
> 1. **Task success:** we discard all trajectories that do not satisfy the task success criterion.
> 2. **Seed diversity:** we enforce the per-seed cap (K = 5) to avoid over-representing any specific layout or biome.
> 3. **Reasoning quality:** for a stratified subset of trajectories, we manually check that the textual reasoning is consistent with the visual scene and the executed actions; trajectories with hallucinated or clearly inconsistent reasoning are removed.
> We have added these concrete numbers and a short description of the three-step filtering in the revision in Appendix A11.
>
> ### **B. (W2) “Notation in Eq. (4): “wait! Let’s re-observe…"**
>
> Thank you for pointing out this confusion. The phrase “wait! Let’s re-observe…” is implemented as a fixed special token prefix following the S1 prompt style [1], and is not meant to distinguish between Y- and Y+. In other words, Y- and Y+ denote the reasoning content itself, while the “wait! Let’s re-observe…” phrase is a constant template prefix.
>
> ### **C. (W3) Meaning of “Number of Samples” and its relation to data efficiency**
> We agree that this connection should be made clearer in the main text.
> As clarified in Appendix A.3, “Num. of samples” in the collecting tasks denotes the number of blocks successfully collected within a fixed interaction horizon (identical time budget / number of environment steps). Thus, for the same time endurance, a higher “number of samples” means that the agent has converted the same amount of interaction into more successful acquisitions, i.e., higher effective data yield from the environment.
>
> ### **D. (W4) Organization of the appendix**
>
> Thank you for this helpful suggestion. The appendix currently begins with a short paragraph “The appendix is organized as follows…”, but we agree that it can be made easier to navigate.
> In the revised version, we have:
> - Turn this paragraph into a bulleted “reading guide”,
> - Highlight the most relevant figures and tables (e.g., ablations, imagined trajectory visualizations, and generalization experiments)
>
>
> ### **E. (W5) Evidence for “strong accuracy and reasoning coherence across categories”**
>
> We apologize for not making the evidence sufficiently explicit at the cited line.
> The claim is supported by Table 2, where the “Score-Stc.” and “Score of Understanding” are computed on the STEVE-21K multimodal QA dataset. These scores are obtained by a GPT-4o-based evaluator that jointly assesses answer correctness and reasoning coherence across different question categories. RIG consistently achieves higher scores than the baselines, indicating both better accuracy and more coherent reasoning chains.
>
> ### **F. (W6–W8) Minor issues (spacing, VPT, figure reference)**
>
> We thank the reviewer for catching these issues. In the revised manuscript:
> - The spacing issue at Ln 1170 has been fixed.
> - The reference to “VPT” at Ln 1190 is corrected to the appropriate method name and citation.
> - The reference at Ln 377 has been updated so that the text correctly refers to the intended figure (Figure 5 in the current version).
>
> [1] Muennighoff, Niklas, et al., s1: Simple test-time scaling, EMNLP 2025.

---

> ### Author Response · Authors · 2025-11-20
> **Comments on Questions**
>
> ### **((Q1) Generalization to other tasks and environments beyond Minecraft**
>
> Our current experiments focus on the Minecraft domain, where we can control environment seeds, biomes, and tasks in a unified way. We view RIG primarily as a general framework for coupling reasoning, imagination, and action, rather than as Minecraft-specific.
> To preliminarily test generalization beyond Minecraft, we continually finetune the trained RIG model on general multimodal understanding data (InternVL-Chat-V1-2-SFT-Data, WikiHow, and WIT). As shown in Table A.4 in the appendix, the resulting model shows strong performance on diverse VQA-style tasks without changing the core architecture, suggesting that the reasoning-imagination coupling transfers to non-Minecraft settings.
> For other embodied environments, we expect that RIG can be instantiated once environment interfaces and data collection pipelines are available. Due to the cost of large-scale data collection in multiple environments, we leave full multi-environment embodied evaluation to future work.
>
> ### **(Q2) Multiple good futures vs. single GT path**
>
> This is an excellent question. In our current data collection, each executed trajectory corresponds to a single “ground-truth” path per episode, generated by a strong teacher policy. However, RIG is not restricted to imagining only one future:
>
> - In S2 / S3, for a given state, we generate **multiple imagined continuations** (rollout 6 successful trajectories on each failure node) using the policy and then label them as positive or negative based on success and returns.
> - The model is trained to score and distinguish among these imagined trajectories, learning that multiple futures are possible but that some are preferable.
>
> Thus, although behavior cloning uses a single executed trajectory per episode, the imagination and review stages expose the model to multiple alternative futures from the same state, and the supervision teaches it to prefer better ones instead of memorizing a single path.
>
> ### **(Q3) Positive trajectories from STEVE-1 vs. negative trajectories from RIG-basic in S3**
> We appreciate the opportunity to clarify this. While STEVE-1’s overall success rate on long-horizon tasks (e.g., 1,500–3,000 steps) is limited (47.4%), it still provides many high-quality short- to mid-horizon segments:
>
> 1. We **decompose long trajectories into shorter sub-trajectories** (e.g., early resource gathering, sub-goals of building), and we treat locally successful segments from STEVE-1 as **positive examples**.
>
> 2. For high-level tasks (e.g., tower building), we further decompose them into simpler low-level tasks (material collection, then structured placement), from which STEVE-1 can still produce a large number of successful traces.
>
> In contrast, RIG-basic is used to generate imagined long-horizon continuations; suboptimal or failed imagined trajectories from RIG-basic naturally serve as negative examples in S3’s dream-review stage.
> Because S3 operates on segments and rollouts around sub-goals, we can collect sufficient positive trajectories from STEVE-1 even when its full-episode success rate is below 100%, and at the same time collect diverse negative examples from RIG-basic. We will add a brief explanation of this decomposition in Section 3.3 and in the appendix.

---

### Official Review · Reviewer_DPLQ · 2025-11-04

**Soundness:** 3
**Presentation:** 2
**Contribution:** 4
**Rating:** 8
**Confidence:** 2

**Summary:**

The paper presents RIG, a unified Transformer-based architecture that jointly learns reasoning, action prediction, and visual imagination for embodied agents. Unlike prior modular systems that combine separate reasoning and world-model components, RIG integrates both abilities end-to-end via a progressive data-collection and training pipeline (S0–S4). The method shows strong empirical gains in data efficiency (17× fewer hours) and performance across embodied control, reasoning, and image-generation benchmarks in the Minecraft environment.

**Strengths:**

1. The explicit synergy between reasoning and imagination in a single model is original and intellectually appealing.
2. The authors conduct extensive experiments showing consistent gains across multiple dimensions (accuracy, sample efficiency, FID/PSNR, and reasoning quality).
3. The staged analysis (Action -> Generation -> Reasoning -> Lookahead) demonstrates how each component contributes to the final performance.
4. The appendix describes data sources, seeds, and prompts, which is appreciated for transparency.

**Weaknesses:**

1. While RIG achieves substantial sample efficiency, it also depends on a large-scale VQA/vision-language model for reasoning annotations and imagination during both training and inference. It remains unclear how much more compute and energy this incurs compared to baselines such as STEVE-1 or MineDreamer. Quantifying this overhead (e.g., FLOPs per inference step, GPU hours, or carbon estimate) would strengthen the argument that RIG is efficient overall.
2. The paper uses GPT-4o for reasoning and reviewing annotations. Although this is explained, it is worth clarifying how much of RIG’s improvement stems from the quality of these annotations versus the architecture itself.
3. The “imagined” trajectories clearly improve performance, but the paper could better illustrate what these visual predictions actually look like and whether they faithfully represent plausible world dynamics rather than generic textures.

**Questions:**

1. How much additional computation does the integrated reasoning–imagination (RIG) pipeline require compared to a pure Dreamer or STEVE-1 baseline?
2. Can the authors provide an estimate of GPU hours or FLOPs per training run?
3. Given that RIG employs a VQA model for annotation and reasoning supervision, how does this affect the inference-time cost when deployed?
4. RIG claims 17× data efficiency, but what is the corresponding energy efficiency?

---

> ### Author Response · Authors · 2025-11-20
> **A. (W1, Q1, Q2, Q4) Compute, energy, and efficiency of RIG**
>
> We thank the reviewer for the thoughtful focus on efficiency. We first clarify that our “17$\times$ efficiency” claim is specifically about sampl efficiency in terms of environment interaction hours (111 h vs. 2000 h), rather than claiming that RIG is already optimal in per-step compute or wall-clock efficiency. In fact, because RIG integrates reasoning, imagination, and lookahead into a single agent, its per-step computation is higher than that of a purely reactive controller. We view improving architectural efficiency (e.g., via model compression, lighter reasoning heads, or shorter imagination horizons) as an important direction for future work.
>
> 1. **Compute per step.** We provide a FLOP analysis of the RIG pipeline. For a typical control step, the policy without visual imagination (Reasoning + Action) requires approximately $2\times 10^{12}$ FLOPs, while the full RIG pipeline with imagination (Reasoning + Action + Prediction) requires about $6\times 10^{12}$ FLOPs. For comparison, the per-step FLOPs of STEVE-1 is about $6\times 10^{10}$, and MineDreamer requires roughly $3\times 10^{14}$ FLOPs per step ($6\times 10^{10}$ for the controller and $3 \times 10^{14}$ for the backend image generator). The detailed compute-cost comparison is listed in Section A.1 and Table A.1 (shown below).
>
> | Method                | Env. h      | Imag. | Reason. | Look. | Summary            | Infer FLOPs        |
> |-----------------------|------------:|:-----:|:-------:|:-----:|--------------------|--------------------|
> | STEVE-1               |  ~2000      |       | ✓       |       | Low-level policy   |  $6 \times 10^{10} $  |
> | MineDreamer           | ~2000* | ✓     |         | ✓     | World-model + RL  | $3 \times 10^{14} $ |
> | RIG-basic             |  **111**    | ✓     | ✓       |       | World-model        | $6 \times 10^{12} $ |
> | RIG-lookahead (Ours)  |  **111**    | ✓     | ✓       | ✓     | World-model        | $6 \times 10^{12} $ |
>
> 2. **Training GPU hours.** Training RIG-basic (without lookahead imagination) requires about 704 GPU hours to train on 111 hours of interaction data, using 64 $\times$ A100 80GB GPUs. The additional lookahead stage for RIG-lookahead requires about 280 GPU hours on the same hardware. We will explicitly report these numbers (and the corresponding per-run configuration) in Section A.2.

---

> ### Author Response · Authors · 2025-11-20
> **B. (W2, Q2, Q3) Role of GPT-4o / VLM supervision vs. RIG architecture**
>
> We appreciate the reviewer’s point about disentangling the contribution of supervision quality from the architecture.
>
> 1. **Use of GPT-4o.** GPT-4o is used only offline in our data pipeline to generate reasoning and review annotations for S2/S3. At inference time, RIG does not call GPT-4o or any external VQA model: the deployed agent is a single model that jointly performs reasoning, action selection, and imagination.
>
> 2. **Attempt with an open-source VLM for annotator.** To assess how far one can go with lighter annotators, we conducted an initial experiment using Qwen3-VL-8B-Instruct as a drop-in replacement for GPT-4o under the same prompts and first-person observations. As detailed in Appendix A.8 and Figure A.7, we observed that:
>   - The model often failed to correctly adopt the first-person viewpoint, producing reasoning that described “the player” from an external perspective instead of the agent’s own view.
>   - It made frequent factual errors about the visual scene, e.g., describing “empty hands” as “holding a wooden pickaxe,” or hallucinating objects that are not present.
>   - The generated reasoning was noticeably noisier and more off-task, with long, redundant text that did not align well with the actual action and goal.
>
> Because these issues led to substantially lower-quality reasoning labels, we decided not to use Qwen3-VL-8B annotations for training RIG in this paper. At the same time, we view these experiments as encouraging evidence that stronger open-source VLMs or student models distilled from GPT-4o could become viable annotators for our pipeline in the near future, improving accessibility for the broader community.
>
> 3. **Architecture vs. supervision quality (Table 1 ablation).** To clarify how much improvement comes from the RIG architecture itself (under the same 111h interaction budget), we report a stage-wise ablation in Table 1:
>
> - ID 0 (Action-only)
>   - Avg. Number of Samples: 7.7 collected blocks
>   - Avg. Accuracy: 8.4% (Dig 9.1, Explore 11.7, Tower 4.4)
> - ID 2 (Action + Reasoning, no visual imagination yet)
>   - Avg. Number of Samples increases to 21.4
>   - Avg. Accuracy rises to 34.6% (Dig 34.2, Explore 31.8, Tower 37.8)
> - ID 4 (Action + Generation + Reasoning + Lookahead, full RIG)
>   - Avg. Number of Samples reaches 80.2 (wood 28.3, grass 137.5, dirt 74.8)
>   - Avg. Accuracy climbs to 79.6% (Dig 65.8, Explore 84.2, Tower 88.7)
>
> The progression from ID 0 → ID 2 → ID 4 shows that:
>
> (a) Adding reasoning supervision (ID 2) on top of the same base policy already yields large gains in both data yield and task success, even though the environment interaction and action labels remain unchanged.
>
> (b) Further adding imagination and lookahead (ID 4) brings additional, consistent improvements across task success, sample efficiency, and generation-related metrics.
>
> Taken together, these results suggest that GPT-4o’s main role is to provide structured and reliable reasoning supervision, while the architecture that tightly couples reasoning, imagination, and action is responsible for a substantial portion of the performance improvement observed in our experiments.

---

> ### Author Response · Authors · 2025-11-20
> **C. (W3, Q3) Faithfulness and interpretation of imagined trajectories**
>
> We agree that it is important to show that the imagined trajectories reflect plausible world dynamics rather than generic textures.
>
> 1. **Qualitative comparison with GPT-4o.** Figures A.5 and A.6 in the appendix have been expanded and clarified to better illustrate this. Given the same instruction “chop a tree,” we compare RIG’s imagined frames and reasoning with those of GPT-4o:
>   - RIG first reasons about which tree is reachable, imagines moving to an appropriate distance, and only then issues the “chop” action. The imagined future frames are geometrically consistent with the actual environment (e.g., tree size and relative position change realistically as the agent moves).
>   - GPT-4o, despite producing visually high-quality images, misjudges the distance to the tree, issues an invalid action, and fails to repair its plan in subsequent steps.
>
> 2. **Lookahead case study and quantitative fidelity.** Figure A.5, A.6 presents a case study focusing on the lookahead module: we show how imagined frames are used to evaluate future outcomes and prune unsafe or ineffective action sequences. This lookahead ability is critical in embodied tasks like fault review or navigation, where accurately estimating distances and affordances in open-world environments is essential.
>
> 3. **Manual audit of imagined trajectories.** To further verify that imagination behaves as expected in practice, we conducted a manual audit of 2,300 randomly sampled imagined sub-trajectories (drawn from rollouts of up to 3,000 iterations). For each sample, we checked whether the imagined frames and textual reasoning were consistent with the underlying task and environment dynamics. Only 22 samples were judged clearly inconsistent (e.g., impossible geometry or reasoning contradicting the visual progression), i.e., <1% of the inspected cases. This manual check provides additional evidence that RIG’s imagined trajectories are generally faithful and reliable for guiding lookahead and self-review.
>
> Beyond qualitative examples, Figure 5 evaluates the fidelity of imagined trajectories quantitatively. Generated frames from RIG are compared to the actual environment frames along the executed trajectory using PSNR and FID, against strong diffusion-based Minecraft generative models with larger parameter counts. RIG achieves better alignment between imagined and real trajectories, indicating that its imagination is not only visually plausible but also dynamically faithful to the underlying environment.
>
> We have revised the appendix to make these points more explicit and to better guide readers through qualitative and quantitative evidence.

---

### Meta-Review · Area_Chair_WdzA · 2026-01-08

**Summary:**

The following summary outlines the reviewers' primary concerns that informed the recommendation for this paper:


+ Dispute over Sample Efficiency Claims: The most significant point of contention involved the "17x sample efficiency" claim. Reviewer JjLn argued that this metric is misleading because it only accounts for the 111 hours of learner-side interaction while ignoring the 2000-hour pre-training cost of the teacher model (STEVE-1) required for data collection.


+ Reliance on High-End VLMs: Reviewers DPLQ and tgFQ expressed concerns regarding the framework's heavy dependency on GPT-4o for generating reasoning and review annotations. This raised questions about the reproducibility and accessibility of the pipeline for researchers without access to such large-scale closed-source models.


+ Methodological Framing: There was a debate regarding whether the work represents a novel, sample-efficient learning paradigm or a complex distillation strategy. Reviewer JjLn contended that the agent is primarily imitating the teacher's actions and GPT-4o's reasoning rather than learning autonomously from environment feedback.


+ Computational and Inference Overhead: Reviewer DPLQ highlighted the potential increase in computational costs (FLOPs per step) and energy consumption incurred by integrating reasoning and imagination into the control loop compared to purely reactive baselines.


+ Domain and Environment Limitations: Multiple reviewers (bvMP, tgFQ) pointed out that the evaluation was restricted to the structured, voxel-based Minecraft environment. They questioned the universality of the conclusions and the method's applicability to continuous action spaces or real-world robotic settings.


+ Fidelity of Imagined Trajectories: A technical concern was raised regarding whether the "imagined" future frames faithfully represent plausible world dynamics or merely generate generic textures.

**Reviewer Concerns:**

**Addressed Concerns**

The following issues were successfully clarified or resolved through the rebuttal and the subsequent updates to the appendix :


+ Inference and Computational Overhead: The authors provided a detailed FLOPs analysis demonstrating that while RIG is more computationally intensive than reactive agents like STEVE-1, it is significantly more efficient than hybrid systems like MineDreamer.


+ Transparency of the Inference Pipeline: The authors elaborated on the "dream-review" process, clarifying how the model uses an internal review head to score imagined trajectories without calling external APIs during inference.


+ Fidelity of Imagination: A manual audit of 2,300 imagined trajectories showed a failure rate of less than 1%, and quantitative PSNR/FID metrics confirmed that RIG’s imagination aligns well with actual environment dynamics.


+ Clarification of "Number of Samples": The authors clarified that this metric refers to the effective data yield (blocks collected) within a fixed interaction horizon, resolving the confusion regarding data efficiency.


**Outstanding Concerns**

Despite the authors' efforts, certain concerns remain partially or fully unaddressed, contributing to the divergence in scores:


+ Inclusion of Teacher Training Costs: While the authors argue that learner-side interaction is the standard metric in distillation literature, the concern remains that the "17x efficiency" claim is context-dependent . Reviewer JjLn maintains that excluding the 2000-hour pre-training cost of the teacher model presents an incomplete picture of the total resource requirement.


+ Dependency on Closed-Source VLMs for Data: Although the authors tested Qwen3-VL as an alternative, they admitted it produced significantly lower-quality reasoning labels compared to GPT-4o . The framework’s current reliance on high-end, closed-source models for its data pipeline remains a barrier to broader community accessibility.


+ Generalization to Non-Voxel Environments: The authors acknowledged that RIG has only been rigorously tested in Minecraft. While they provided a theoretical argument for applicability to continuous control, the lack of empirical evidence in simulators like CARLA or MetaWorld leaves the claim of "universal generalist policy" under-supported.


+ Integration with Reinforcement Learning: The current system uses Rejection Sampling Fine-Tuning (RFT) rather than on-policy RL. Reviewers noted that the potential for imagination to directly optimize action strategies via policy gradients remains unexplored in the current manuscript.

**Reviewer Scores:**

Reviewer DPLQ (Score: 8): This reviewer would likely have maintained their score of 8, as the rebuttal provided the requested quantitative FLOPs analysis and confirmed the high fidelity of imagined trajectories through a manual audit.


Reviewer bvMP (Score: 6): They would likely have maintained their post-rebuttal score of 6, as the authors successfully clarified the "number of samples" metric and the specifics of the data filtering process.


Reviewer tgFQ (Score: 6): This reviewer might have increased their score to 7, given their statement that almost all doubts regarding annotation costs and future reinforcement learning integration were resolved.


Reviewer JjLn (Score: 4): They would likely have maintained their score of 4, as the fundamental conceptual disagreement regarding the "hidden" cost of teacher model training remains a significant barrier.

---

### Decision · Program_Chairs · 2026-01-26

Accept (Poster)